# Fitting summary statistics of neural data with a differentiable spiking network simulator

Guillaume Bellec*, Shuqi Wang*, Alireza Modirshanechi, Johanni Brea◇, Wulfram Gerstner◇

Laboratory of Computational Neuroscience
École polytechnique fédérale de Lausanne (EPFL)
`first.lastname@epfl.ch`

## Abstract

Fitting network models to neural activity is an important tool in neuroscience. A popular approach is to model a brain area with a probabilistic recurrent spiking network whose parameters maximize the likelihood of the recorded activity. Although this is widely used, we show that the resulting model does not produce realistic neural activity. To correct for this, we suggest to augment the log-likelihood with terms that measure the dissimilarity between simulated and recorded activity. This dissimilarity is defined via summary statistics commonly used in neuroscience and the optimization is efficient because it relies on back-propagation through the stochastically simulated spike trains. We analyze this method theoretically and show empirically that it generates more realistic activity statistics. We find that it improves upon other fitting algorithms for spiking network models like GLMs (Generalized Linear Models) which do not usually rely on back-propagation. This new fitting algorithm also enables the consideration of hidden neurons which is otherwise notoriously hard, and we show that it can be crucial when trying to infer the network connectivity from spike recordings.

## 1 Introduction

Modelling neural recordings has been a fundamental tool to advance our understanding of the brain. It is now possible to fit recurrent spiking neural networks (RSNNs) to recorded spiking activity [1, 2, 3, 4, 5]. The resulting network models are used to study neural properties [6, 7, 8, 9] or to reconstruct the anatomical circuitry of biological neural networks [4, 5, 10].

Traditionally a biological RSNN is modelled using a specific Generalized Linear Model [11] (GLM) often referred to as the Spike Response Model (SRM) [12, 11, 13]. The parameters of this RSNN model are fitted to data with the maximum likelihood estimator (MLE). The MLE is consistent, meaning that if the amount of recorded data becomes infinite it converges to the true network parameters when they exist. However, when fitting neural activity in practice the MLE solutions are often reported to generate unrealistic activity [14, 15, 16] showing that this method is not perfect despite it's popularity in neuroscience. We also find that these unrealistic solution emerge more easily when hidden neurons outside of the recorded units have a substantial impact on the recorded neurons. This is particularly problematic because the likelihood is not tractable with hidden neurons and it raises the need for new methods to tackle the problem.

To address this, we optimize *sample-and-measure* loss functions in addition to the likelihood: these loss functions require *sampling* spiking data from the model and *measuring* the dissimilarity between recorded and simulated data. To measure this dissimilarity we suggest to compare summary statistics

---

*Equal contributions. ◇ Senior authors.

popular in neuroscience like the peristimulus time histogram (PSTH) and the noise-correlation (NC). Without hidden neurons, this method constrains the network to generate realistic neural activity but without biasing the MLE solution in the theoretical limit of infinite data. In practice, it leads to network models generating more realistic activity than the MLE. With hidden neurons, the sample-and-measure loss functions can be approximated efficiently whereas the likelihood function is intractable. Although recovering the exact network connectivity from the recorded spikes remains difficult [10], we show on artificial data that modelling hidden neurons in this way is crucial to recover the connectivity parameters. In comparison, methods like MLE which ignore the hidden activity wrongly estimate the connectivity matrix.

In practice the method is simple to optimize with automatic differentiation but there were theoretical and technical barriers which have prevented earlier attempts. The first necessary component is to design an efficient implementation of back-propagation in stochastic RSNN inspired by straight-through gradient estimators [17, 18] and numerical tricks from deterministic RSNNs [19]. Previous generative models of spiking activity relying on back-propagation used artificial neural networks [20, 21] which are not interpretable model in terms of connectivity and neural dynamics. Previous attempts to include hidden neurons in RSNN models did not rely on back-prop but relied on expectation maximization [11, 22] or reinforce-style gradients [23, 24, 25, 26] which have a higher variance [27]. There exist other methods to fit neural data using back-propagation and deep learning frameworks but they do not back-propagate through the RSNN simulator itself, rather they require to engineer and train a separate deep network to estimate a posterior distribution [28, 29, 30, 31] or as a GAN discriminator [32, 33, 21]. The absence of a discriminator in the sample-and-measure loss function connects it with other simple generative techniques used outside of the context of neural data [34, 35, 36].

Our implementation of the algorithm is published online openly [2].

## 2   A recurrent spiking neural network (RSNN) model

We will compare different fitting techniques using datasets of spiking neural activity. We denote a tensor of $K^{\mathcal{D}}$ recorded spike trains as $\boldsymbol{z}^{\mathcal{D}} \in \{0, 1\}^{K^{\mathcal{D}} \times T \times n_{\mathcal{V}}}$ where $n_{\mathcal{V}}$ is the total number of visible neurons recorded simultaneously and $T$ is the number of time steps. To model the biological network which produced that activity, we consider a simple model that can capture the recurrent interactions between neurons and the intrinsic dynamics of each neuron. This recurrent network contains $n_{\mathcal{V}+\mathcal{H}}$ neurons connected arbitrarily and split into a visible and a hidden population of sizes $n_{\mathcal{V}}$ and $n_{\mathcal{H}}$. Similarly to [1, 4, 37, 14] we use a GLM where each unit is modelled with a SRM neuron [12] with $u_{t,j}$ being the distance to the threshold of neuron $j$ and its spike $z_{t,j}$ is sampled at time step $t$ from a Bernoulli distribution $\mathcal{B}$ with mean $\sigma(u_{t,j})$ where $\sigma$ is the sigmoid function. The dynamics of the stochastic recurrent spiking neural network (RSNN) are described by:

$$z_{t,j} \quad \sim \quad \mathcal{B}\left(\sigma(u_{t,j})\right) \qquad \text{with} \qquad u_{t,j} = \frac{v_{t,j} - v_{\text{thr}}}{v_{\text{thr}}} \tag{1}$$

$$v_{t,j} \quad = \quad \sum_{i=1}^{n_{\mathcal{V}+\mathcal{H}}} \sum_{d=1}^{d_{\max}} W_{j,i}^d z_{t-d,i} + b_j + \mathcal{C}_{t,j} \,, \tag{2}$$

where $\boldsymbol{W}$ defines the spike-history and coupling filters spanning $d_{\max}$ time-bins, $\boldsymbol{b}$ defines the biases, $v_{\text{thr}} = 0.4$ is a constant, and $\mathcal{C}$ is a spatio-temporal stimulus filter processing a few movie frames and implemented here as a convolutional neural network (CNN) (this improves the fit accuracy as seen in [14, 38] and in Figure S1 from the appendix). Equations (1) and (2) define the probability $\mathcal{P}_\phi(\boldsymbol{z})$ of simulating the spike trains $\boldsymbol{z}$ with this model and $\phi$ represents the concatenation of all the network parameters ($\boldsymbol{W}$, $\boldsymbol{b}$ and the CNN parameters). Traditionally the parameters which best explain the data are given by the MLE: $\text{argmax}_{\phi}\mathcal{P}_\phi(\boldsymbol{z}^{\mathcal{D}})$ [1, 7, 37, 5, 4]. When all neurons are visible, the likelihood factorizes as $\prod_t \mathcal{P}_\phi(\boldsymbol{z}_t^{\mathcal{D}}|\boldsymbol{z}_{1:t-1}^{\mathcal{D}})$, therefore the log-likelihood can be written as the negative cross-entropy ($CE$) between $\boldsymbol{z}_t^{\mathcal{D}}$ and $\sigma(\boldsymbol{u}_t^{\mathcal{D}}) = \mathcal{P}_\phi(\boldsymbol{z}_t^{\mathcal{D}} = \boldsymbol{1}|\boldsymbol{z}_{1:t-1}^{\mathcal{D}})$ where $\boldsymbol{u}_t^{\mathcal{D}}$ is computed as in equation (2) with $\boldsymbol{z} = \boldsymbol{z}^{\mathcal{D}}$. So when all neurons are recorded and $\boldsymbol{z}^{\mathcal{D}}$ are provided in the dataset the computation of the MLE never needs to simulate spikes from the model and it is sufficient to minimize the loss function:

$$\mathcal{L}_{MLE} = -\log \mathcal{P}_\phi(\boldsymbol{z}^{\mathcal{D}}) = CE(\boldsymbol{z}^{\mathcal{D}}, \sigma(\boldsymbol{u}^{\mathcal{D}})) \,. \tag{3}$$

---

[2]Code repository: `https://github.com/EPFL-LCN/pub-bellec-wang-2021-sample-and-measure`

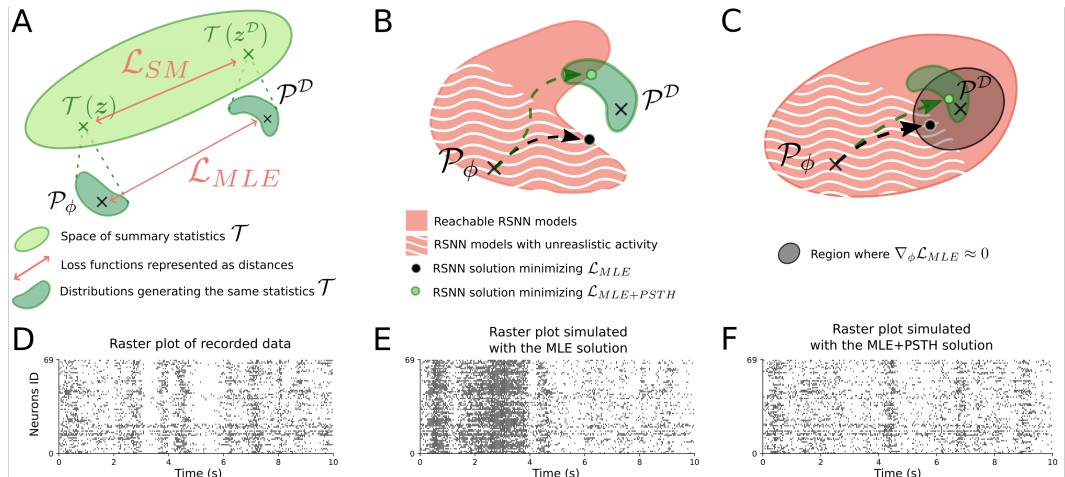

Figure 1: **A)** The distribution $\mathcal{P}_\phi$ represents the RSNN model and $\mathcal{P}_\mathcal{D}$ represents the true biological network, the goal is to bring $\mathcal{P}_\phi$ close to $\mathcal{P}_\mathcal{D}$. The loss function $\mathcal{L}_{MLE}$ is represented as the distance between $\mathcal{P}_\phi$ and $\mathcal{P}_\mathcal{D}$ because it is equal up to a constant to $D_{KL}(\mathcal{P}_\mathcal{D}, \mathcal{P}_\phi)$. We draw the space of summary statistics $\mathcal{T}$ to represent the loss function $\mathcal{L}_{PSTH}$ as the distance between the statistics $\mathcal{T}(z)$ simulated from the RSNN model $\mathcal{P}_\phi$ and measured from the data $\mathcal{P}_\mathcal{D}$. **B)** Even if the model is misspecified and $\mathcal{P}_\phi$ cannot perfectly match the true distribution $\mathcal{P}_\mathcal{D}$, when $\mathcal{L}_{MLE+PSTH}$ is minimized the statistics $\mathcal{T}$ are indistinguishable between simulation and data (the green dot lands in the dark green area). When minimizing MLE alone, the solution (black dot) might generate unrealistic activity. **C)** When the RSNN model is expressive enough to represent the true distribution but the data is insufficient, so there is some uncertainty about the true network parameters (black ellipse): minimizing $\mathcal{L}_{MLE+PSTH}$ favours solutions with realistic PSTH. **D)** Spikes recorded simultaneously from the V1-dataset [39]. **E)** Simulated spiking response from an RSNN minimizing $\mathcal{L}_{MLE}$. **F)** Same as **E** but with $\mathcal{L}_{MLE+PSTH}$.

## 3 The sample-and-measure loss functions

In this section we describe the sample-and-measure loss functions which include simulated data in a differentiable fashion in the optimization objective, a direct benefit is to enable the consideration of hidden neurons. We define the *sample-and-measure* loss functions as those which require *sampling* spike trains $z \in \{0,1\}^{K \times T \times n_\mathcal{V}+\mathcal{H}}$ from the model $\mathcal{P}_\phi$ and *measuring* the dissimilarity between the recorded and simulated data. This dissimilarity is defined using some statistics $\mathcal{T}(z)$ and the generic form of the sample-and-measure loss functions is:

$$\mathcal{L}_{SM} = d\Big( \mathcal{T}(z^\mathcal{D}), \mathbb{E}_{\mathcal{P}_\phi}[\mathcal{T}(z)] \Big), \tag{4}$$

where $d$ is a dissimilarity function, like the mean-squared error or the cross entropy. To compute the expectations $\mathbb{E}_{\mathcal{P}_\phi}$ we use Monte-Carlo estimates from a batch of simulated trials $z$. For example to match the PSTH between the simulated and recorded data, we consider the statistics $\mathcal{T}(z)_{t,i} = \frac{1}{K} \sum_k z_{t,i}^k$ and evaluate the expectation with the unbiased estimate $\bar{\sigma}_{t,i} = \frac{1}{K} \sum_k \sigma(u_{i,t}^k)$. Denoting the PSTH of the data as $\bar{z}_{t,i}^\mathcal{D} = \frac{1}{K} \sum_k z_{t,i}^{k,\mathcal{D}}$ and choosing $d$ to be the cross-entropy, we define the sample-and-measure loss function for the PSTH:

$$\mathcal{L}_{\text{PSTH}} = CE(\bar{z}^\mathcal{D}, \bar{\sigma}). \tag{5}$$

When all neurons are visible, we minimize the loss function $\mathcal{L}_{MLE+SM} = \mu_{MLE}\mathcal{L}_{MLE} + \mu_{SM}\mathcal{L}_{SM}$ where $\mu_{MLE}, \mu_{SM} > 0$ are constant scalars. When there are hidden neurons, the log-likelihood is intractable. Instead we minimize the negative of a lower bound of the log-likelihood (see appendix D for a derivation inspired by [40, 23, 24, 25]):

$$\mathcal{L}_{ELBO} = CE(z^\mathcal{D}, \sigma(u^\mathcal{V})), \tag{6}$$

with $\sigma(u^\mathcal{V})$ being the firing probability of the visible neurons, where the visible spikes $z^\mathcal{V}$ are clamped to the recorded data $z^\mathcal{D}$ and the hidden spikes $z^\mathcal{H}$ are sampled according to the model

dynamics. Hence the implementation of $\mathcal{L}_{ELBO}$ and $\mathcal{L}_{SM}$ are very similar with the difference that the samples used in $\mathcal{L}_{SM}$ are not clamped (but all our results about $\mathcal{L}_{SM}$ are also valid when they are clamped if we use the extended definition given in Appendix D). To compute the gradients with respect to these loss functions we use back-propagation which requires the propagation of gradients through the stochastic samples $z$. If they were continuous random variables, one could use the reparametrization trick [27], but extending this to discrete distributions is harder [17, 18, 41, 42, 43]. One way to approximate these gradients is to relax the discrete dynamics into continuous ones [41] or to use relaxed control variates [42], but we expect that the relaxed approximations become more distant from the true spiking dynamics as the network architecture gets very deep or if the network is recurrent. Instead, we choose to simulate the exact spiking activity in the forward pass and use straight-through gradient estimates [17, 18] in the backward pass by defining a pseudo-derivative $\frac{\partial z_{t,i}}{\partial u_{t,i}}$ over the binary random variables $z_{t,i} \sim \mathcal{B}(\sigma(u_{t,i}))$. We use here the same pseudo-derivative $\frac{\partial z_{t,i}}{\partial u_{t,i}} = \gamma \max(0, 1 - |u_{t,i}|)$ as in deterministic RNNs [19] because the dampening factor (here $\gamma = 0.3$) can avoid the explosive accumulation of approximation errors through the recurrent dynamics [44]. Although the resulting gradients are biased, they work well in practice.

**A geometrical description of a sample-and-measure loss function**  In the remaining paragraphs of this section we provide a geometrical representation and a mathematical analysis of the loss function $\mathcal{L}_{SM}$. For this purpose, we consider that the recorded spike trains $z^{\mathcal{D}}$ are sampled from an unknown distribution $\mathcal{P}_{\mathcal{D}}$ and we formalize that our goal is to bring the distribution $\mathcal{P}_\phi$ as close as possible to $\mathcal{P}_{\mathcal{D}}$. In this view, we re-write $\mathcal{L}_{SM} = d(\mathbb{E}_{\mathcal{P}_{\mathcal{D}}}[\mathcal{T}(z)], \mathbb{E}_{\mathcal{P}_\phi}[\mathcal{T}(z)])$ and we re-interpret $\mathcal{L}_{MLE}$ as the Kullback-Leibler divergence ($D_{KL}$) from $\mathcal{P}_\phi$ to $\mathcal{P}_{\mathcal{D}}$. This is equivalent because the divergence $D_{KL}(\mathcal{P}_{\mathcal{D}}, \mathcal{P}_\phi)$ is equal to $-\mathbb{E}_{\mathcal{P}_{\mathcal{D}}}[\log \mathcal{P}_\phi(z^{\mathcal{D}})]$ up to a constant.

In Figure 1 we represent the losses $\mathcal{L}_{MLE}$ and $\mathcal{L}_{PSTH}$ in the space of distributions and we can represent $\mathcal{L}_{MLE} = D_{KL}(\mathcal{P}_{\mathcal{D}}, \mathcal{P}_\phi)$ as the distance between $\mathcal{P}_{\mathcal{D}}$ and $\mathcal{P}_\phi$. To represent the sample-and-measure loss function $\mathcal{L}_{SM}$ (or specifically $\mathcal{L}_{PSTH}$ in Figure 1), we project the two distributions onto the space of summary statistics $\mathcal{T}$ represented in light green. Hence, these projections represent the expected statistics $\mathbb{E}_{\mathcal{P}_\phi}[\mathcal{T}(z)]$ and $\mathbb{E}_{\mathcal{P}_{\mathcal{D}}}[\mathcal{T}(z^{\mathcal{D}})]$ and $\mathcal{L}_{SM}$ can be represented as the distance between the two projected statistics.

Although minimizing $\mathcal{L}_{MLE}$ should recover the true distributions (i.e. the biological network) if the recorded data is sufficient and the model is well specified, these ideal conditions do not seem to apply in practice because the MLE solution often generates unrealistic activity (see Figure 1D-E). Panels B and C in Figure 1 use the geometrical representation of panel A to summarize the two main scenarios where minimizing $\mathcal{L}_{MLE+SM}$ is better than $\mathcal{L}_{MLE}$ alone. In panel B, we describe a first scenario in which the model is misspecified meaning that it is not possible to find $\phi^*$ so that $\mathcal{P}_{\mathcal{D}} = \mathcal{P}_{\phi^*}$. In this misspecified setting, there is no guarantee that the MLE solution yields truthful activity statistics and it can explain why the MLE solution generates unrealistic activity (Figure 1E). In this case, adding a sample-and-measure loss function can penalize unrealistic solutions to solve this problem (Figure 1B and F). Another possible scenario is sketched in panel C. It describes the case where the model is well specified but $\mathcal{L}_{MLE}$ is flat around $\mathcal{P}_{\mathcal{D}}$ for instance because too few trials are recorded or some neurons are not recorded at all. In that case we suggest to minimize $\mathcal{L}_{MLE+SM}$ to nudge the solution towards another optimum where $\mathcal{L}_{MLE}$ is similarly low but the statistics $\mathcal{T}$ match precisely. In this sense, $\mathcal{L}_{SM}$ can act similarly as a Bayesian log-prior to prefer solutions producing truthful activity statistics.

**Theoretical analysis of the sample-and-measure loss function**  To describe formal properties of the sample-and-measure loss function $\mathcal{L}_{SM}$, we say that two distributions are indistinguishable according to the statistics $\mathcal{T}$ if the expectation $\mathbb{E}[\mathcal{T}(z)]$ is the same for both distributions. We assume that the dissimilarity function $d(\mathcal{T}, \mathcal{T}')$ reaches a minimum if and only if $\mathcal{T} = \mathcal{T}'$ (this is true for the mean-squared error and the cross-entropy). Then for any statistics $\mathcal{T}$ and associated dissimilarity function $d$ we have:

**Property 1.** *If the RSNN model is expressive enough so that there exists parameters $\phi^\circ$ for which $\mathcal{P}_\phi$ and $\mathcal{P}^{\mathcal{D}}$ are indistinguishable according to the statistics $\mathcal{T}$, then $\phi^\circ$ is a global minimum of $\mathcal{L}_{SM}$. Reciprocally, if this minimum is reached then $\mathcal{P}_\phi$ and $\mathcal{P}^{\mathcal{D}}$ are indistinguishable according to $\mathcal{T}$.*

This property is a direct consequence of our assumption on the function $d$. If $\mathcal{T}$ measures the PSTH it means that the optimized simulator produces the same PSTH as measured in the data. This can be true even if the model is misspecified which is why we represented in Figure 1B that the RSNN minimizing $\mathcal{L}_{MLE+PSTH}$ lands in the dark green region where the PSTH of the data is matched accurately. As one may also want to target other statistics like the noise correlation (NC), it is tempting to consider different statistics $\mathcal{T}_1$ and $\mathcal{T}_2$ with corresponding dissimilarity functions $d_1$ and $d_2$ and to minimize the sum of the two losses $\mathcal{L}_{SM_1+SM_2} = \mu_1 \mathcal{L}_{SM_1} + \mu_2 \mathcal{L}_{SM_2}$ where $\mu_1, \mu_2 > 0$ are constant scalars. Indeed if $d_1$ and $d_2$ follow the same assumption as previously, we have (see Figure 2 for an illustration):

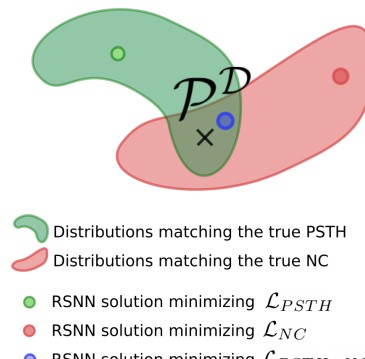

Distributions matching the true PSTH

Distributions matching the true NC

● RSNN solution minimizing $\mathcal{L}_{PSTH}$

● RSNN solution minimizing $\mathcal{L}_{NC}$

● RSNN solution minimizing $\mathcal{L}_{PSTH+NC}$

Figure 2: Combining $\mathcal{L}_{PSTH}$ and $\mathcal{L}_{NC}$.

**Property 2.** *If the RSNN model is expressive enough so that there exists $\phi^\circ$ for which $\mathcal{P}_\mathcal{D}$ and $\mathcal{P}_\phi$ are indistinguishable according to both statistics $\mathcal{T}_1$ and $\mathcal{T}_2$, then $\phi^\circ$ is a global minimum for $\mathcal{L}_{SM_1+SM_2}$. Reciprocally, if this minimum is reached, $\mathcal{P}_\mathcal{D}$ and $\mathcal{P}_\phi$ are indistinguishable according to $\mathcal{T}_1$ and $\mathcal{T}_2$.*

This is again a direct consequence of the assumptions on $d_1$ and $d_2$. Additionally Figure 1C conveys the idea that $\mathcal{L}_{SM}$ and $\mathcal{L}_{MLE}$ are complementary and $\mathcal{L}_{SM}$ can be interpreted as a log-prior. This interpretation is justified by the following Property which is similar to an essential Property of Bayesian log-priors. It shows that when it is guaranteed to recover the true model by minimizing $\mathcal{L}_{MLE}$, minimizing the regularized likelihood $\mathcal{L}_{MLE+SM}$ will also recover the true model.

**Property 3.** *If the RSNN is well specified and identifiable so that $\mathcal{P}_\mathcal{D} = \mathcal{P}_{\phi^*}$ and in the limit of infinite data, then the global minimum of $\mathcal{L}_{MLE+SM}$ exists, it is unique and equal to $\phi^*$.*

To prove this, we first note that all the conditions are met for the consistency of MLE so $\phi^*$ is the unique global minimum of $\mathcal{L}_{MLE}$. Also the assumption $\mathcal{P}_\mathcal{D} = \mathcal{P}_{\phi^*}$ is stronger than the assumption required in Properties 1 and 2 (previously $\phi^\circ$ only needed to match summary statistics: with parameters $\phi^*$ the model is indeed matching any statistics) so it is also a global minimum of $\mathcal{L}_{SM}$. As a consequence it provides a global minimum for the summed loss $\mathcal{L}_{MLE+SM}$. This solution is also unique because it has to minimize $\mathcal{L}_{MLE}$ which has a unique global minimum.

It may seem that those properties are weaker than the classical properties of GLMs: in particular $\mathcal{L}_{SM+MLE}$ is not convex anymore because of $\mathcal{L}_{SM}$ and the optimization process is not guaranteed to find the global minimum. This could be a disadvantage for $\mathcal{L}_{MLE+SM}$ but we never seemed to encounter this issue in practice. In fact, as we argue later when analyzing our simulation results, the optimum of $\mathcal{L}_{MLE+SM}$ found empirically always seem to be closer to the biological network than the global minimum of $\mathcal{L}_{MLE}$. We think that it happens because the conditions for the consistency of the MLE and Property 3 (identifiability and infinite data) are not fully met in practice. On the contrary, the Properties 1 and 2 hold with a misspecified model or a limited amount of recorded data which may explain the success of the sample-and-measure loss functions in practice. See Figure 1 for a geometrical interpretation.

## 4 Numerical simulations without hidden neurons

For our first quantitative results we consider a single session of in-vivo recordings from the primary visual cortex of a monkey watching repetitions of the same natural movie [39]. We refer to this dataset as the V1-dataset. It contains the spike trains of 69 simultaneously recorded neurons for 120 repetitions lasting 30 seconds. We only keep the first 80 repetitions in our training set and 10 repetitions are used for early-stopping. Performances are tested on the remaining 30. In our first numerical results we do not include hidden neurons.

To illustrate that minimizing $\mathcal{L}_{MLE}$ alone does not fit well the statistics of interest, we show in Figure 3A-C the learning curves obtained when minimizing $\mathcal{L}_{MLE}$, $\mathcal{L}_{PSTH}$ and $\mathcal{L}_{MLE+PSTH}$. We evaluate the PSTH correlation between simulated and recorded activity every time the training loss function reaches a new minimum. With MLE in Figure 3B, the PSTH correlation saturates at a sub-optimal level and drops unexpectedly when $\mathcal{L}_{MLE}$ decreases. In contrast, with the sample-and-measure loss function, the PSTH correlation improves monotonously and steadily (see Figure 3A).

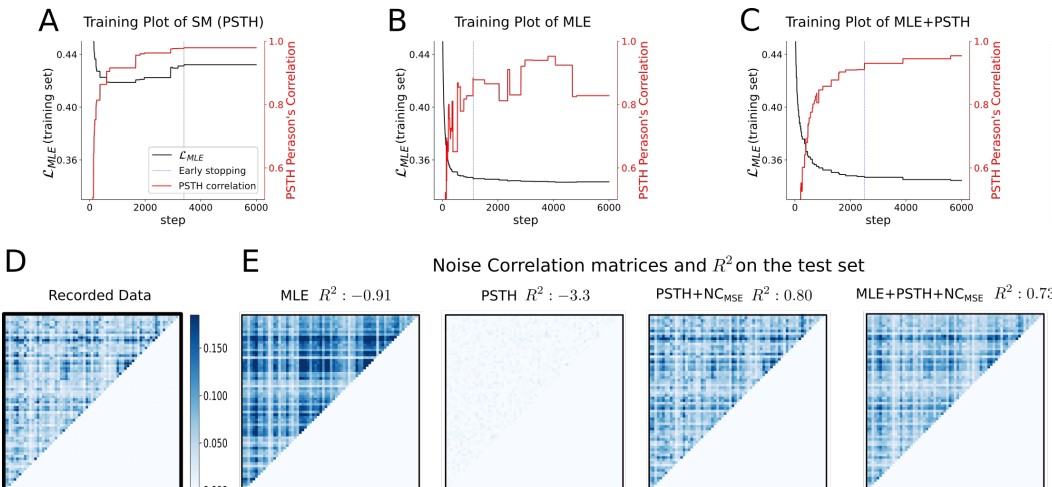

Figure 3: Learning curves and NC performance summary on the V1-dataset. **A-C)** Negative log-likelihood (i.e. $\mathcal{L}_{MLE}$) and PSTH correlation on the training set for three networks trained with $\mathcal{L}_{PSTH}$, $\mathcal{L}_{MLE}$ and $\mathcal{L}_{MLE+PSTH}$. The red curves represent the correlation between the PSTH of the recorded and simulated data. To show how the training loss influences the resulting PSTH correlation we plot a new point of the red curve only when the training loss reaches a new minimum. The vertical blue line represents the best network achieving the lowest validation losses (i.e. $\mathcal{L}_{PSTH}$, $\mathcal{L}_{MLE}$ and $\mathcal{L}_{MLE+PSTH}$ for the three corresponding plots). **D)** Noise-correlation (NC) matrix as recorded in the data. x- and y-axis represent the neuron identities. **E)** NC matrices when the spikes are simulated from the model, it uses the same colorbar as in **D)**. $R^2$ values are reported to compare the recorded NC and the simulated ones, more results are provided in Table 1.

In Figure 3C, one sees that minimizing $\mathcal{L}_{MLE+SM}$ produces low values of $\mathcal{L}_{MLE}$ and maximizes efficiently the PSTH correlation.

We then fit simultaneously the PSTH and the noise-correlation (NC) on the V1-dataset. The NC matrix is complementary to the PSTH and it is used regularly to measure the fit performance [21, 37, 14]. Its entries can be viewed as a measure of functional connectivity, and each coefficient is defined for the neuron pair $i$ and $j$ as the correlation of their activity. Concretely it is proportional to the statistics $\mathcal{T}(\boldsymbol{z})_{i,j} = \frac{1}{KT} \sum_{k,t} (z_{t,i}^k - \bar{z}_{t,i})(z_{t,j}^k - \bar{z}_{t,j})$ where $\bar{z}_{t,i}$ is the PSTH (see appendix C for details). Therefore the natural sample-and-measure loss function for NC is the mean-squared error between the coefficients $\mathcal{T}(\boldsymbol{z}^{\mathcal{D}})_{i,j}$ and the Monte-carlo estimates $\frac{1}{KT} \sum_{k,t} (\sigma(u_{t,i}^k) - \bar{\sigma}_{t,i})(\sigma(u_{t,j}^k) - \bar{\sigma}_{t,j})$. We denote the resulting loss function as $\mathcal{L}_{NC_{MSE}}$. We also tested an alternative loss $\mathcal{L}_{NC}$ which uses the cross entropy instead of mean-squared error and compares: $\mathcal{T}(\boldsymbol{z}^{\mathcal{D}})_{i,j} = \frac{1}{KT} \sum_{k,t} z_{t,i}^{k,\mathcal{D}} z_{t,j}^{k,\mathcal{D}}$ with the Monte-Carlo estimate $\frac{1}{KT} \sum_{k,t} \sigma(u_{t,i}^k) \sigma(u_{t,j}^k)$.

We compare quantitatively the effects of the loss functions $\mathcal{L}_{MLE}$, $\mathcal{L}_{PSTH}$, $\mathcal{L}_{NC}$ and $\mathcal{L}_{NC_{MSE}}$ and their combinations on the V1-dataset. The results are summarized in Table 1 and NC matrices are shown in Figure 3E. The network fitted solely with $\mathcal{L}_{PSTH}$ shows the highest PSTH correlation while its noise correlation is almost zero everywhere (see Figure 3E), but this is corrected when adding $\mathcal{L}_{NC}$ or $\mathcal{L}_{NC_{MSE}}$. In fact a network minimizing $\mathcal{L}_{MLE}$ alone yields lower performance than minimizing $\mathcal{L}_{PSTH+NC}$ for both metrics. When combining all losses into $\mathcal{L}_{MLE+PSTH+NC}$ or $\mathcal{L}_{MLE+PSTH+NC_{MSE}}$ the log-likelihood on the test set is not compromised and it fits better the PSTH and the NC: the coefficient of determination $R^2$ of the NC matrix improves by a large margin in comparison with the MLE solution. Analyzing the failure of MLE we observe in Figure 3 that the NC coefficients are overestimated. We wondered if the fit was mainly impaired by trials with unrealistically high activity as in Figure 1E. But that does not seem to be the case, because the fit remains low with MLE ($R^2 = -0.78$) even when we discard trials where the firing probability of a neuron is higher than $0.85$ for 10 consecutive time steps.

We report in Figure S2 the PSTH correlation and noise-correlation in a different format to enable a qualitative comparison with the results obtained with a spike-GAN on the same dataset (see

Table 1: Performance summary on the test set when fitting RSNN models to the V1-dataset. The precise definition of the performance metrics are given in Appendix C. The standard deviation across neurons is provided for the PSTH correlation. The variability across different network initialization is relatively low in comparison with the difference across algorithms, for instance we computed the standard deviation of $\mathcal{L}_{MLE}$ over 3 seeds for MLE and MLE+PSTH+NC$_{MSE}$ and found respectively $2 \cdot 10^{-5}$ and $3 \cdot 10^{-4}$. For the noise correlation $R^2$, the standard deviation was $9 \cdot 10^{-3}$ for MLE+PSTH+NC$_{MSE}$.

| Method | PSTH correlation | Noise Correlation $(R^2)$ | Negative log-likelihood $\mathcal{L}_{MLE}$ (on test set) |
|---|---|---|---|
| MLE | $0.67 \pm 0.16$ | $-0.91$ | $0.370$ |
| PSTH | $0.72 \pm 0.15$ | $-3.3$ | $0.44$ |
| PSTH+NC$_{MSE}$ | $0.69 \pm 0.15$ | $0.80$ | $0.50$ |
| MLE+PSTH+NC$_{MSE}$ | $0.69 \pm 0.15$ | $0.73$ | $0.370$ |

Figure 3B,C from [21]). The fit is qualitatively similar even if we do not need a separate discriminator network. Also our RSNN model is better suited to make interpretations about the underlying circuitry because it models explicitly the neural dynamics and the recurrent interactions between the neurons (whereas a generic stochastic binary CNN without recurrent connections was used with the spike-GAN).

We also compare our approach with the 2-step method which is a contemporary alternative to MLE for fitting RSNNs [14]. The PSTH and noise correlation obtained with the 2-step method were measured on another dataset of 25 neurons recorded in-vitro in the retina of the Rat [9]. We trained our method on the same dataset under the two stimulus conditions and a quantitative comparison is summarized in Table S6. Under a moving bar stimulus condition we achieve a higher noise correlation (3% increase) and a higher PSTH correlation (19% increase). But this difference might be explained by the use of a linear-simulus filter [14] instead of a CNN. Under a checkerboard stimulus condition, the 2-step method was tested with a CNN but we still achieve a better noise-correlation (5% improvement) with a slightly worse PSTH correlation (2% decrease). Another difference is that it is not clear how the 2-step method can be extended to model the activity of hidden neurons as done in the following section.

In summary, this section shows that using a differentiable simulator and simple sample-and-measure loss functions leads to a competitive generative model of neural activity. The approach can also be generalized to fit single-trial statistics as explained in the Appendix D and Figure S3.

## 5   Model identification

Beyond simulating realistic activity statistics, we want the RSNN parameters to reflect a truthful anatomical circuitry or realistic neural properties. To test this, we consider a synthetic dataset generated by a *target network* for which we know all the parameters. We build this target network by fitting it to the V1-dataset and sample from this model a synthetic dataset of similar size as the V1-dataset (80 training trials of approximately 30 seconds). Since our target network can generate as much data as we want, we simulate a larger test set of 480 trials and a larger validation set of 40 trials. We then fit *student networks* on this synthetic dataset and compare the parameters $\phi$ of the *student* networks with the ground-truth parameters $\phi^*$ of the *target* networks.

**Well specified model without hidden neurons**   As a first comparison we consider the simplest case where the target network is fully observed: the target network consist of 69 visible neurons and each student network is of the same size. This is in fact the ideal setting where the log-likelihood is tractable and the MLE enjoys strong theoretical guarantees. In particular if the CNN weights are not trained and are copied from the target-network, the loss function $\mathcal{L}_{MLE}$ is convex with respect to the remaining RSNN parameters $\phi$ and the target network is identifiable [45]. The resulting fitting performance is summarized in Figure 4A where we show the NC matrix and the connectivity matrix $\left(\sum_d W_{i,j}^d\right)$ for the target network and two students networks. We do not show the PSTH because all methods already fit it well on the V1-dataset (see Table 1). In this setting, combining $\mathcal{L}_{NC}$ and

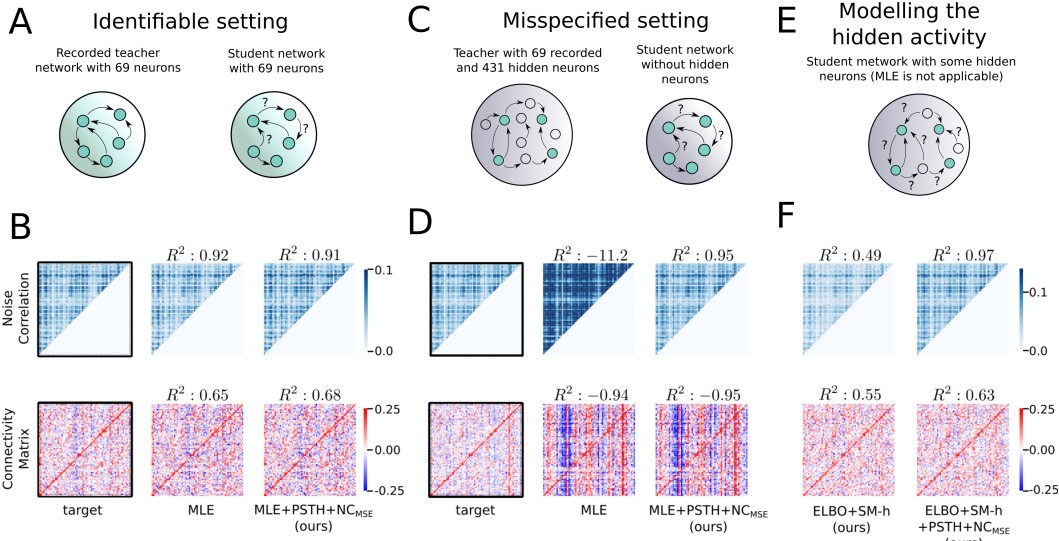

Figure 4: **A, B**) Results of the model identification experiment in the fully identifiable setting. **A** summarizes the simulation setup. The results are shown in **B**: in the first row, we show the noise correlation matrix. In the second row, we show the connectivity matrix $\sum_d W_{ji}^d$ where the x- axis indicates pre-synaptic neuron $i$ and y-axis indicates post-synaptic neuron $j$. **C, D**) Same as **A, B** but in a misspecified setting: the target network has $500$ neurons and student networks have $69$ neurons. **E, F**) Same as before with the same target network as in **C, D**, but the student has some hidden neurons. When averaging $R^2$ of the connectivity matrices over 5 different student network initialization, we find $R^2 = 0.56 \pm 0.0069$ for ELBO+SM-h and $0.63 \pm 0.013$ for ELBO+SM-h+PSTH+NC$_{MSE}$. For the noise correlation, we find $0.66 \pm 0.1$ for ELBO+SM-h and $0.95 \pm 0.2$ for ELBO+SM-h+PSTH+NC$_{MSE}$. Here the number of hidden neurons in the student network is the same as in the target network, but this is not necessary as seen in Table S4. The connectivity matrices are displayed with higher resolution in Figure S5 and S6.

$\mathcal{L}_{PSTH}$ with $\mathcal{L}_{MLE}$ brings almost no advantage: the MLE already provides a good reconstruction of the NC and connectivity matrices.

**Model misspecification when ignoring hidden neurons**   From these results we hypothesize that this fully identifiable setting does not capture the failure of MLE observed with real data because the recorded neurons are embedded in a much larger biological network that we cannot record from. To model this, we construct another synthetic dataset based on a larger target network of $500$ neurons where the first $69$ neurons are fitted to the neurons recorded in the V1-dataset and the remaining $431$ are only regularized to produce a realistic mean firing rate (see appendix for simulation details). As in the standard setting where one ignores the presence of hidden neurons, we first consider that the student networks model only the first $69$ visible neurons. This model is therefore misspecified because the number of neurons are different in the target and student networks, hence this setting is well described by the scenario sketched in Figure 1B.

The results are shown in Figure 4B. We found that MLE is much worse than the sample-and-measure method in this misspecified setting and the results resemble better what has been observed with real data. With MLE the noise-correlation coefficient are over estimated and the overall fit is rather poor (negative $R^2$), but it significantly improves after adding the sample-and-measure loss functions ($R^2 = 0.95$). This suggest that ignoring the impact of hidden neurons can explain the failure of MLE experienced in the real V1-dataset. We find little relationship between the student and teacher connectivity matrices (only the connectivity between visible neurons are compared, see Figure 3). This suggests that the standard strategy, where the hidden neurons are ignored, is unlikely to be informative about true cortical connectivity.

**Well specified model with hidden neurons**   To investigate whether including hidden neurons leads to more truthful network models, we take the same target network of $500$ neurons and fit now student

networks of the same size (500 neurons) but where only the first 69 are considered visible (Figure 4C). Since the model is well specified but data about the hidden neurons is missing, this experiment is well summarized by the scenario of Figure 1C. We use $\mathcal{L}_{ELBO}$ for the visible units and we add a sample-and-measure loss function $\mathcal{L}_{SM-h}$ to constrain the average firing rate of the hidden neurons which are completely unconstrained otherwise (see appendix). As seen in Figure 4C, it yields more accurate NC matrix ($R^2 = 0.49$) and connectivity matrix ($R^2 = 0.55$) compared to the previous misspecified models which did not include the hidden neurons. When we add sample-and-measure loss functions to fit the PSTH and NC of the visible neurons, the noise-correlation matrix and connectivity matrix are fitted even better ($R^2 = 0.97$ and $R^2 = 0.63$). Quantitatively, the $R^2$ for the connectivity matrices are almost as high as in the easy setting of panel A where the target network is fully-visible and identifiable. Although the student network had exactly the same number of hidden neurons in Figure 4 E-F, the success is not dependent on the exact number of hidden neurons as shown in Table S4. Rather, assuming a small hidden population size of only 10 neurons was enough to alleviate the failure observed in the absence of hidden neurons in Figure 4 C-D. Table S4 also shows however that the accuracy of the reconstruction improves substantially if the hidden population is made larger in the student network.

## 6 Discussion

We have introduced the sample-and-measure method for fitting an RSNN to spike train recordings. This method leverages deep learning software and back-propagation for stochastic RSNNs to minimize sample-and-measure loss functions. A decisive feature of this method is to model simply and efficiently the activity of hidden neurons. We have shown that this is important to reconstruct trustworthy connectivity matrices in cortical areas. We believe that our approach paves the way towards better models with neuroscientifically informed biases to reproduce accurately the recorded activity and functional connectivity. Although we have focused here on GLMs, PSTH and NC, the method is applicable to other spiking neuron models and statistics.

**Perspective** One of the promising aspects of our method is to fit models which are much larger. One way to do this, is to combine neurons from separate sessions in a single larger network by considering them alternatively visible or hidden. This problem was tackled partially in [46, 47]. It is natural to implement this with our method and we believe that it is a decisive step to produce models with a dense coverage of the recorded areas.

To investigate if our method is viable in this regime we carried out a prospective scaling experiment on a dataset from the Mouse brain recorded with multiple Neuropixels probes across 58 sessions [48]. The goal of this scaling experiment is only to evaluate the amount of computing resources required to fit large networks. We ran three fitting experiments with 2, 10 and 20 sessions respectively. Focusing on neurons from the visual cortices, it yielded models with 527, 2219 and 4995 neurons respectively. Each simulation could be run on a single NVIDIA V100 GPU and running 100 training epochs took approximately 4, 12 and 36 hours respectively. We conclude that this large simulation paradigm is approachable with methods like ours and we leave the fine-tuning of these experiments and the analysis of the results for future work.

## Acknowledgments and Disclosure of Funding

This research was supported by Swiss National Science Foundation (no. 200020_184615) and the Intel Neuromorphic Research Lab. Many thanks to Christos Sourmpis, Gabriel Mahuas, Ulisse Ferrari, Franz Scherr and Wolfgang Maass for helpful discussions. Special thanks to Stéphane Deny, Olivier Marre and Ulisse Ferrari for sharing with us the Retina dataset and to Matthew Smith and Adam Kohn for making their dataset publicly available.

**Authors contributions** GB and SW conceived the project initially. SW did most of the simulations under the supervision of GB. All authors contributed significantly to the theory and the writing.

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
