# Appendices of:

# Fitting summary statistics of neural data with a differentiable spiking network simulator

## A  Datasets

**V1-dataset**  The dataset we used was collected by Smith and Kohn [49] and is publicly available at: http://crcns.org/data-sets/vc/pvc-11. In summary, macaque monkeys were anesthetized with Utah arrays placed in the primary visual cortex (V1). In our analysis, we considered population spiking activity of monkey-I in response to a gray-scale natural movie. The movie is about a monkey wading through water. It lasts for 30 seconds (with sampling rate 25Hz) and was played repeatedly for 120 times. Similarly as in [21], we used the last 26 seconds of the movies and recordings. Each frame of the movie has $320 \times 320$ pixels and we downsampled them to $27 \times 27$ pixels. We used the recording from the 69 neurons with time bins 40ms and considered that there cannot be more than one spike per bin ($5\%$ of the time bins had more than one spike).

**Synthetic dataset**  Two target networks are trained using the V1-dataset: one with no hidden neuron and one with 431 hidden neurons which makes 500 neurons in total. To build the target network without hidden neurons, we fitted a network with the loss function $\mathcal{L}_{MLE+PSTH+NC}$. For the target network with hidden neurons, we train a network using $\mathcal{L}_{MLE+SM-h+PSTH+NC}$.

**Retina dataset**  The data we used were the same as [14] and was initially published in [9]. It was generously shared with us privately. It contained recorded spike trains for 25 OFF Alpha retinal ganglion cells' in the form of binarized spike counts in 1.667ms bins. There were two stimulus conditions. For the checkerboard, the unrepeated movie (1080s) plus one repeated movie (600s in total for 120 repetitions) were used for training and the other repeated movie (480s in total for 120 repetitions) were used for testing. For the moving bar, the unrepeated movie (1800s) plus one repeated movie (166s in total for 50 repetitions) were used for training and the other repeated movie (322s in total for 50 repetitions) were used for testing.

## B  Simulation details

**For the V1- and synthetic datasets**  The model combines a spatio-temporal CNN and an RSNN. Input to the CNN consists of 10 consecutive movie frames. The CNN has 2 hidden layers and its output is fed into the RSNN. To feed the images to the CNN the 10 gray-scaled images are concatenated on the channel dimension. The two hidden layers include convolution with 16 and 32 filters, size 7 by 7 (with padding) followed by a ReLU activation function and then a MaxPool layer with kernel size 3 and stride 2 as in [50]. The weights from the CNN to the RSNN are initialized with a truncated normal distribution with standard deviation $\frac{1}{\sqrt{n_{in}}}$ where $n_{in}$ in the number of inputs in the weight matrix. The tensor of recurrent weights $\boldsymbol{W}$ consider spike history of last 9 frames ($d_{\max}$) and the weight distribution is initialized as a truncated normal distribution with standard deviation $\frac{1}{\sqrt{d_{\max} n_{\mathcal{H}+\mathcal{V}}}}$. The bias $\boldsymbol{b}$ is initialized with zero. The voltage threshold $v_{\mathrm{thr}}$ is set to 0.4 and the dampening factor $\gamma$ is 0.3. We used an Adam optimizer. More hyper-parameters like learning rates are given in Table S2 and Table S3. To implement the loss $\mathcal{L}_{MLE+PSTH+NC}$, we process the CNN once and simulate the RSNN twice. Once the RSNN is clamped to the recorded spikes to compute $\mathcal{L}_{MLE}$ or $\mathcal{L}_{ELBO}$, the second time the sample and generated "freely" to compute $\mathcal{L}_{PSTH}$ and $\mathcal{L}_{NC}$.

**For the retina dataset experiment**  Since the time step is much smaller for the Retina dataset than for the V1-dataset (1.67ms rather than 40ms) the temporal filters have to be larger to take into account the full temporal context. For both the receptive fields of the CNN and the tensor $\boldsymbol{W}$ we chose to cover time scales that are consistent with [14]. Hence we adapted the model architecture from the previous paragraph and added as a first layer of the CNN a causal temporal convolution (Conv1D with appropriate padding). The temporal convolution has a receptive filed of 300 time bins and outputs 16 filters. In the RSNN we choose $d_{\max} = 24$ so that the spike history filter covers around 40ms. Two fitting algorithms were tested, one with $\mathcal{L}_{MLE}$ and the other one with $\mathcal{L}_{MLE+single-trial+NC}$. The loss function $\mathcal{L}_{single-trial}$ is used to fit single-trial statistics as defined in Appendix D and we used it here to replace $\mathcal{L}_{PSTH}$ because some movies of training dataset are unrepeated and we saw in Figure S3 that it fits the PSTH almost as well as $\mathcal{L}_{PSTH}$. To implement the

loss $\mathcal{L}_{MLE+single-trial+NC}$, we process the CNN once and simulate the RSNN twice for $T$ time steps. The first time the RSNN is clamped to the recorded spikes for $T_{gt}$ time steps and then clamping is terminated and the RSNN generates samples "freely" for the next $T - T_{gt}$ time steps. For the first $T_{gt}$ time steps, $\mathcal{L}_{MLE}$ is computed. And for the rest $T - T_{gt}$ time steps where the activity is not clamped, $\mathcal{L}_{single-trial}$ is computed as the cross entropy between $\mathbf{z}^{\mathcal{D}}$ and the spike probabilities. We also run the RSNN a second time with the same CNN input and without any clamping to compute $\mathcal{L}_{NC}$. For each gradient descent step, we sample uniformly from the dataset a batch of size $K^{\mathcal{D}} = K_m \times K_t$ gathering truncated movie clips and corresponding spikes with $K_m$ different starting time points and from $K_t$ different movies. The hyper-parameters can be found in Table S5.

## C  Performance metrics

For the definition of our performance metrics we use the following notations. The trial averaged firing probability of neuron $i$ in the time bin $t$ is denoted $\bar{z}_{t,i} = \frac{1}{K}\sum_k z_{t,i}^k$ where $z_{t,i}^k \in \{0,1\}$ is the spike and $K$ is the number of trials. Neuron $i$'s mean firing rate is further computed as $\bar{z}_i = \frac{1}{T}\sum_t \bar{z}_{t,i}$ where $T$ is the number of time steps.

**Peristimulus time histogram (PSTH) correlation**   The fit performance of the PSTH is measured by the Pearson's correlation between the simulated PSTH and the recorded PSTH. Hence for each neuron the PSTH correlation is defined by:

$$\rho_i^{\text{PSTH}} = \frac{\sum_t \left(\bar{z}_{t,i} - \bar{z}_i\right)\left(\bar{z}_{t,i}^{\mathcal{D}} - \bar{z}_i^{\mathcal{D}}\right)}{\sqrt{\sum_t \left(\bar{z}_{t,i} - \bar{z}_i\right)^2}\sqrt{\sum_t \left(\bar{z}_{t,i}^{\mathcal{D}} - \bar{z}_i^{\mathcal{D}}\right)^2}} \, , \tag{7}$$

and a slightly better estimator of the asymptotical Pearson correlation which is less noisy can be estimated by replacing $\bar{z}_{t,i}$ and $\bar{z}_i$ with $\bar{\sigma}_{t,i}$ and $\bar{\sigma}_i$.

**Noise-correlation matrix**   Pairwise noise correlations are computed as in [46]. We first define total covariance $M_{i,j}^{\text{total}}$ and noise covariance $M_{i,j}^{\text{noise}}$ between neuron $i$ and $j$.

$$M_{i,j}^{\text{total}} \;\; = \;\; \frac{1}{TK}\sum_{t,k}\left(z_{t,i}^k - \bar{z}_i\right)\left(z_{t,j}^k - \bar{z}_j\right) \tag{8}$$

$$M_{i,j}^{\text{noise}} \;\; = \;\; \frac{1}{TK}\sum_{t,k}\left(z_{t,i}^k - \bar{z}_{t,i}\right)\left(z_{t,j}^k - \bar{z}_{t,j}\right) \tag{9}$$

Then in the performance tables we report the normalized noise correlation $\mathcal{M}_{i,j}^{\text{noise}}$ for $i \neq j$:

$$\mathcal{M}_{i,j}^{\text{noise}} = \frac{M_{i,j}^{\text{noise}}}{\sqrt{M_{i,i}^{\text{total}}M_{j,j}^{\text{total}}}} \, . \tag{10}$$

We then define the coefficient of determination of the NC matrix $R^2$ as in [14]. Given the NC matrices computed from the data $\mathcal{M}_{i,j}^{\text{noise},\mathcal{D}}$ and the NC matrix obtained from the simulation $\mathcal{M}_{i,j}^{\text{noise},\phi}$ we define $\mathcal{M}^{\text{noise},\mathcal{D}} = \frac{1}{n_{\mathcal{V}}^2}\sum_{i,j}\mathcal{M}_{i,j}^{\text{noise},\mathcal{D}}$ and:

$$R^2 = 1 - \frac{\sum_{i,j}\left(\mathcal{M}_{i,j}^{\text{noise},\mathcal{D}} - \mathcal{M}_{i,j}^{\text{noise},\phi}\right)^2}{\sum_{i,j}\left(\mathcal{M}_{i,j}^{\text{noise},\mathcal{D}} - \mathcal{M}^{\text{noise},\mathcal{D}}\right)^2} \, . \tag{11}$$

## D  Derivations of the loss functions

**Normalization of the sample-and-measure functions**   Most sample-and-measure may be defined one multiplicative constant away from their formal definition. For instance when computing $\mathcal{L}_{MLE}$ we compute the binary cross entropy between the relevant probabilities aggregate them by taking the mean and not the sum. We find the resulting number to be easier to interpret because is it independent from the number of trials and the number of time steps.

**Noise correlation**   We tested two sample-and-measure loss function for the noise correlation. We explain here why the Monte-Carlo estimate of the simulated statistics is unbiased for $\mathcal{L}_{NC}$ but the same argument applies to $\mathcal{L}_{NC_{MSE}}$.

We consider the statistics $\mathcal{T}(\mathbf{z})_{ij} = \frac{1}{KT}\sum_{t,k} z_{t,i}^k z_{t,j}^k$ which measure the frequency of coincident spikes between neurons $i$ and $j$. Since $z_{t,i}^k$ and $z_{t,j}^k$ are independent given the past, we have $\mathbb{E}_{\mathcal{P}_\phi}\left[z_{t,i}^k z_{t,j}^k\right] =$

$\mathbb{E}_{\mathcal{P}_\phi}\left[\sigma(u_{t,i}^k)\sigma(u_{t,j}^k)\right]$ so we use the following Monte-Carlo estimate $\pi_{i,j}^\phi = \frac{1}{KT}\sum_{t,k}\sigma(u_{t,i}^k)\sigma(u_{t,j}^k)$ to evaluate the expected simulated statistics in equation (4). Choosing the dissimilarity $d$ to be the cross entropy and denoting $\pi_{i,j}^{\mathcal{D}} = \mathcal{T}(z^{\mathcal{D}})_{ij}$ we define:

$$\mathcal{L}_{NC} = \sum_{i,j} CE(\pi_{i,j}^{\mathcal{D}}, \pi_{i,j}^\phi) \tag{12}$$

As an attempt to replace the terms in $\mathcal{L}_{NC_{MSE}}$ which take into account the correlation from the PSTH, we tried to add a related correction term in $\mathcal{L}_{NC}$. To do do we considerd another loss $\mathcal{L}_{NC\text{-shuffled}}$ which is computed like $\mathcal{L}_{NC}$ but where we shuffle the trial identities in $z_i$ and not in $z_j$. It seems that it was not as efficient as $\mathcal{L}_{NC_{MSE}}$.

**Single-trial statistics**  Since both PSTH and NC are trial-averaged statistics we wondered whether another simple measuring model could account for single-trial statistics. We therefore considered the following problem which is notoriously challenging for the MLE [15]: we clamp the network to the recorded data until time $t$ and generate a simulated spike train for $t' > t$. With MLE the network activity quickly diverges away from the real data. To measure this quantitatively we estimate the multi-step log-likelihood $\mathcal{P}_\phi(z_{t+\Delta t}^{\mathcal{D}}|z_0^{\mathcal{D}}\cdots z_t^{\mathcal{D}})$. It is intractable but an unbiased Monte-Carlo estimate can be computed. The multi-step log-likelihood drops quickly as $\Delta t$ increases as expected for MLE in Figure S3.

To resolve that issue, we first suggest an extension of the definition of $\mathcal{L}_{SM}$ in equation 4 which formalizes the clamping condition:

$$\mathcal{L}_{SM} = d\big(\mathbb{E}_{\mathcal{P}_{\mathcal{D}}}\left[\mathcal{T}(z)\mid c\right], \mathbb{E}_{\mathcal{P}_\phi}\left[\mathcal{T}(z)\mid c\right]\big), \tag{13}$$

where we have introduced a condition $c$ into the expectations. All the theory and the geometrical interpretations can be extended with this conditioning, but this allows to formalize that the visible units can be clamped to the recorded data. For instance if we choose $c$ such that $z_{1:t}^{\mathcal{V}} = z_{1:t}^{\mathcal{D}}$ we formalize a sample-and-measure loss function for which the visible units are clamped until time $t$.

Back to the problem of fitting the multi-step log-likelihood, we consider the sample-and-measure loss function where $\mathcal{T}$ is identity, $\sigma(u)$ is the Monte-Carlo estimator and $d$ is the cross-entropy. It yields:

$$\mathcal{L}_{single\text{-}trial} = CE(z^{\mathcal{D}}, \sigma(u^{\mathcal{V}})), \tag{14}$$

which is pretty much computed like $\mathcal{L}_{MLE}$ but where the data is only clamped until time $t$. Note that since the statistics $\mathcal{T}$ do not involve a trial average, the computation of the expectation is not very precise but it may be improved for the expectation $\mathbb{E}_{\mathcal{P}_\phi}$ by averaging over multiple batches clamped to the same data. Although this is an interesting direction we did not try it and always sample a single batch per clamping condition. When using this loss function, we see in Figure S3B that MLE only better just at the first time step after the clamping terminates and optimizing $\mathcal{L}_{single\text{-}trial}$ makes better prediction after that. To provide a meaningful baseline we show the m-step likelihood obtained with a theoretical model fitting perfectly the PSTH without being aware of the clamping history. The multi-step likelihood obtained with $\mathcal{L}_{single\text{-}trial}$ is above this baseline for 5 time-steps (200ms) on the training set proving that the model tries to make a clever usage of the trial specific firing history up to this duration.

**Derivation of the ELBO**  Like for capturing single trial statistics, the most natural way to fit neural activity in the presence of hidden neurons is to minimize the cross-entropy between the visible spikes and their probability while sampling from the hidden neurons. Here we want to show that this is actually the negative of a variational lower bound of the maximum likelihood. Following [40], for any distribution $q(z^{\mathcal{H}})$ of the hidden neural activity we have:

$$\log \mathcal{P}_\phi(z^{\mathcal{D}}) = \log \sum_{z^{\mathcal{H}}} \mathcal{P}_\phi(z^{\mathcal{D}}, z^{\mathcal{H}}) \tag{15}$$

$$= \log \sum_{z^{\mathcal{H}}} q(z^{\mathcal{H}}) \frac{\mathcal{P}_\phi(z^{\mathcal{D}}, z^{\mathcal{H}})}{q(z^{\mathcal{H}})} \tag{16}$$

$$\geq \sum_{z^{\mathcal{H}}} q(z^{\mathcal{H}}) \log \frac{\mathcal{P}_\phi(z^{\mathcal{D}}, z^{\mathcal{H}})}{q(z^{\mathcal{H}})} \tag{17}$$

Writing $z_t$ as the concatenation of $z_t^{\mathcal{D}}$ and $z_t^{\mathcal{H}}$, we now choose specifically $q$ so that for all $t$: $q(z_t^{\mathcal{H}}) = \mathcal{P}_\phi(z_t^{\mathcal{H}}|z_{1:t-1})$, using the factorization and seeing that the probability factorizes as follows: $\mathcal{P}_\phi(z^{\mathcal{D}}, z^{\mathcal{H}}) = \prod_t \mathcal{P}_\phi(z_t^{\mathcal{D}}, z_t^{\mathcal{H}}|z_{1:t-1}) = \prod_t \mathcal{P}_\phi(z_t^{\mathcal{D}}|z_{1:t-1})\cdot\prod_t \mathcal{P}_\phi(z_t^{\mathcal{H}}|z_{1:t-1})$, some products inside the log are cancelling out and we found the lower bound:

$$\log \mathcal{P}_\phi(z^{\mathcal{D}}) \geq \mathbb{E}_q\left[\sum_t \log \mathcal{P}_\phi(z_t^{\mathcal{D}}|z_{1:t-1})\right] \tag{18}$$

$$= -\mathbb{E}_q\left[CE(z^{\mathcal{D}}, \sigma(u^{\mathcal{D}}))\right], \tag{19}$$

| Method | learning rate | batch size | $\mu_{PSTH}$ | $\mu_{NC}$ | $\mu_{MLE}$ |
|---|---|---|---|---|---|
| MLE | | | 0 | 0 | 1 |
| PSTH | 1e-3 | 20 | 1 | 0 | 0 |
| MLE+PSTH | | | 0.5 | 0.5 | 0 |
| PSTH+NC | | | 0.11 | 0.89 | 0 |
| MLE+PSTH+NC | | | 0.1 | 0.5 | 0.4 |
| MLE+PSTH+NC$_{MSE}$ | | | 0.1 | 50 | 0.4 |

Table S2: Hyper-parameter table used when fitting the V1-dataset (Figure 3).

| Method | learning rate | batch size | $\mu_{PSTH}$ | $\mu_{NC}$ | $\mu_{MLE}$ | $\mu_{SM-h}$ |
|---|---|---|---|---|---|---|
| MLE | | | 0 | 0 | 1 | 0 |
| MLE+PSTH+NC | 1.5e-3 | 20 | 0.1 | 0.7 | 0.2 | 0 |
| MLE+SM-h | | | 0 | 0 | 0 | 1e-3 |
| MLE+SM-h+PSTH+NC | | | 0.1 | 0.7 | 0.2 | 1e-3 |

Table S3: Hyper-parameter table used when fitting the synthetic dataset (Figure 4). Early-stopping was used on 40 validation trials to prevent over-fitting.

Interestingly, a similar loss function can also be formulated as a sample-and-measure loss function. To do so we consider the definition from equation (13) with the condition $c$ being $z^{\mathcal{V}} = z^{\mathcal{D}}$ meaning that all the visible units are clamped to the data. Choosing otherwise $\mathcal{T}$ to be the identity and $d$ as the cross-entropy, we obtain the following loss function denoted as $\mathcal{L}_{ELBO-SM}$:

$$\mathcal{L}_{ELBO-SM} = CE\left(z^{\mathcal{D}}, \mathbb{E}_{\mathcal{P}_\phi}\left[\sigma(u^{\mathcal{D}})\right]\right). \tag{20}$$

Comparing the two loss functions we see that the essential difference is the placement of the expectation $\mathbb{E}_{\mathcal{P}_\phi}$. In practice our current optimization minimizes (19) rather than (20) because we sample a single trial for each clamping condition and apply stochastic gradient descent with momentum. This implements implicitly the averaging of the gradients which corresponds better to the expectation from equation (19). However it is also possible to minimize (20) by averaging the Monte-Carlo estimates obtained with multiple simulations with the same clamping condition. With enough sample it may provides a better estimate of the expectation $\mathbb{E}_{\mathcal{P}_\phi}\left[\sigma(u^{\mathcal{D}})\right]$. The down side of this alternative is that it requires to sample more RSNN trajectories for each gradient update which may consume compute time inefficiently. On the other hand, this might be relevant in another setting or at the end of training to benefit from the theoretical properties of the sample-and-measure loss function. We leave this to future work.

**Regularization of the firing rate of hidden neurons**   When simulating hidden neurons which are never recorded it is desirable to insert that as much prior knowledge as possible about the hidden activity to keep the network model in a realistic regime. The most basic prior is to assume that every neuron $i$ should have a realistic average firing rate, to implement this we design again a sample-and-measure objective as a variant of $\mathcal{L}_{PSTH}$. This time we consider that the statistics $\mathcal{T}$ are the average firing rate of a neuron $\mathcal{T}(z_i) = \sum_{t,k} z_{t,i}^k$. This results in the objective $\mathcal{L}_{SM-h}$ which is otherwise similar to $\mathcal{L}_{PSTH}$ as defined in equation (5). Unfortunately the objective cannot be implemented as such because of one missing element: the empirical probability $\pi_i^{\mathcal{D}}$ of a hidden neuron. Instead we simply take another neuron $j$ at random in the visible population and use this average firing rate in place of the probability $\pi_i^{\mathcal{D}}$. In this way, the distribution of average firing rates across neurons of the hidden neurons is realistic at a population level because it becomes the same in the recorded population and in simulated population.

| Number of Hidden Neurons | Noise Correlation $R^2$ | Connectivity Matrix $R^2$ |
|---|---|---|
| 0 | 0.95 | $-0.95$ |
| 10 | 0.96 | 0.50 |
| 200 | 0.97 | 0.59 |
| 400 | 0.95 | 0.62 |

Table S4: Performance summary on the test set when fitting RSNN models with variable number of hidden neurons to the synthetic dataset. ELBO+SM-h+PSTH+NC$_{MSE}$ method is used when hidden neurons are included, and MLE+PSTH+NC$_{MSE}$ method is used when no hidden neurons are included.

| Dataset | Method | $T_{gt}$ | $T$ | $K_m$ | $K_t$ | learning rate |
|---|---|---|---|---|---|---|
| Moving bars stimulus | MLE + single-trial + NC | 45 | 50 | 20 | 8 | 5e-3 |
| | MLE | 50 | 50 | 20 | 8 | 5e-3 |
| Checkerboard stimulus | MLE + single-trial + NC | 75 | 80 | 30 | 1 | 1e-3 |
| | MLE | 75 | 75 | 8 | 4 | 1e-3 |

Table S5: Hyper-parameter table for fitting the Retina Dataset, the definition of the hyper-parameter is given in Appendix B. The results are reported in Table S6.

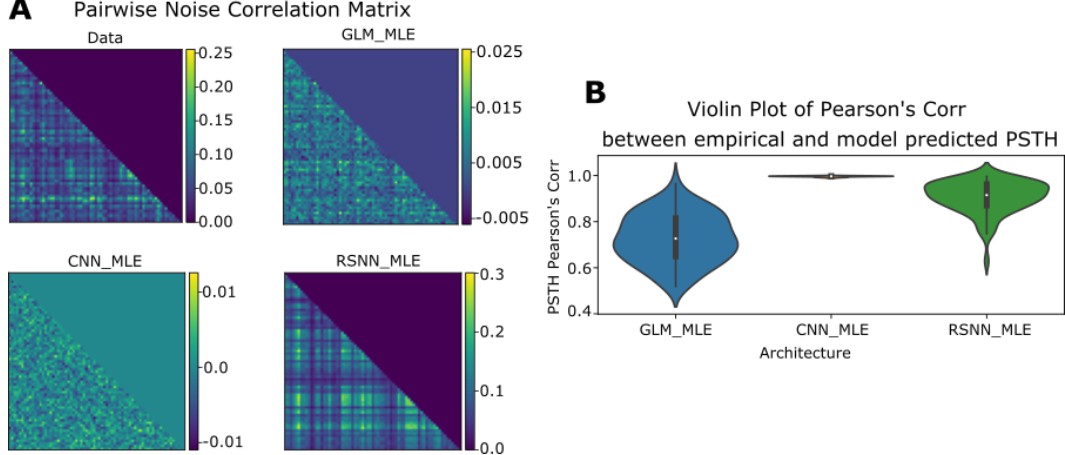

Figure S1: **Comparison of the network architecture** As a preliminary experiment we compared our network architecture against the official GLM code [1]. In our architecture a CNN replaces the spatio-temporal stimulus filter of the CNN, both models are fitted with MLE. **A** Noise correlation matrix of the different models. We included a control architecture where we pruned out the recurrent connection. It is called CNN because only the CNN parameters become relevant. **B** The PSTH correlation computed on the training set. The violon plot represents the distribution of neurons.

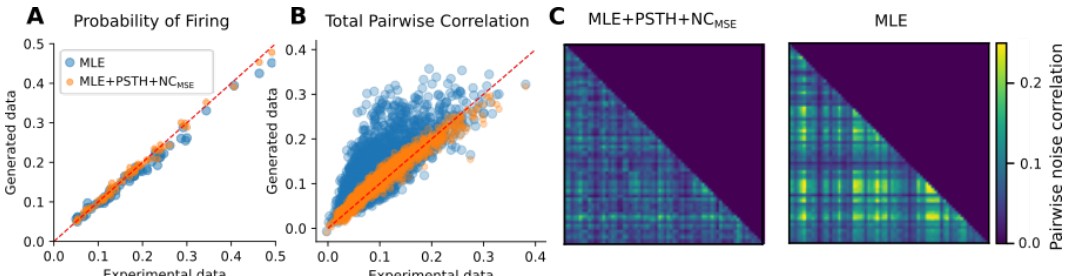

Figure S2: **Comparison with SpikeGAN** This figure is meant to be compared with the Figure 3 from [21]. In this other paper, the authors fitted a spike-GAN to the same dataset. We argue that the PTSH correlation and NC coefficients are as good qualitatively as the results obtained in [21].

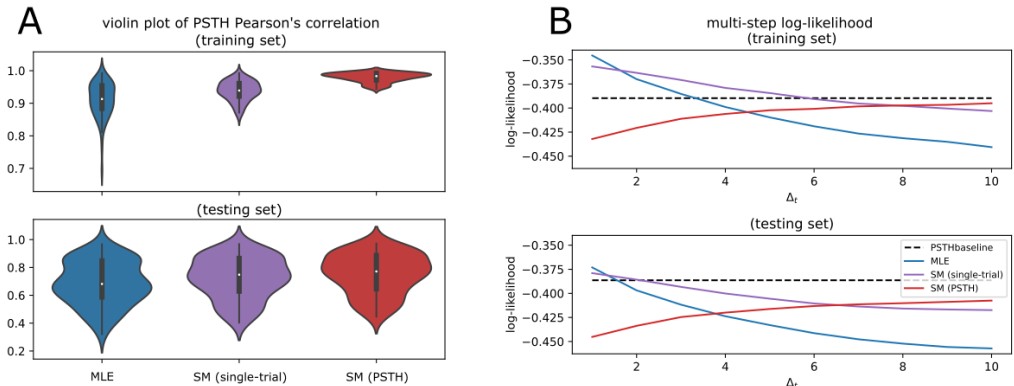

Figure S3: **Improving the mutli-step log-likelihood** We tackle the challenge identified in [15]. We evaluated the multi-step log-likelihood $\log \mathcal{P}_\phi(z_{t+\Delta t}^{\mathcal{D}}|z_0^{\mathcal{D}} \cdots z_t^{\mathcal{D}})$ as explained in the main text, and we trained two networks to minimize $\mathcal{L}_{MLE}$ and $\mathcal{L}_{single-trial}$ respectively. **A**) The PSTH correlation of the different models trained in this context. **B**) The multi-step log-likelihood is reported for different models. The dashed baseline represent the ideal model which would always fire a spike with the true PSTH probability. The blue baseline is $\mathcal{L}_{MLE}$ it has never seen self-generated activity during the training, so it's performance drops quickly when the network is not clamped anymore ($\Delta t > 0$). The red-baseline is a model trained with $\mathcal{L}_{PSTH}$ only, it is increasing because the model has never been clamped during training. Therefore it is not trained to be accurate right after the clamping terminates.

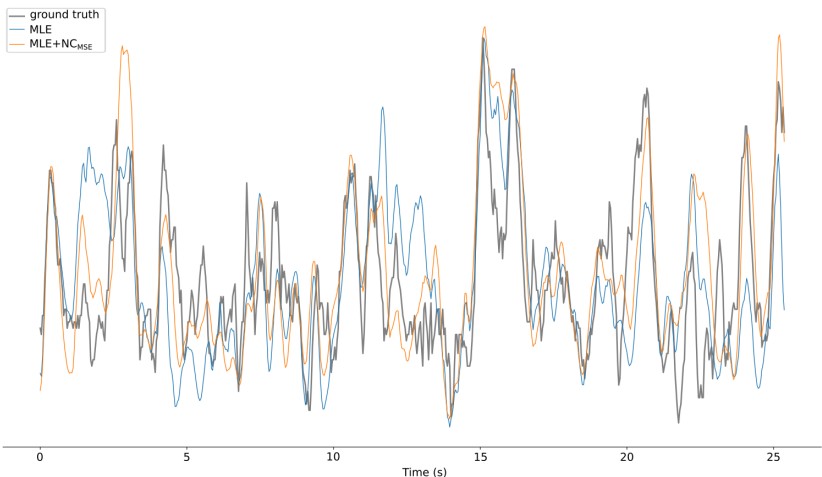

Figure S4: Example of PSTH obtained for a neuron from the V1-dataset. The PSTH are smoothed with a window size of 280ms.

| Method | Moving bars stimulus | | Checkerboard stimulus | |
|---|---|---|---|---|
| | PSTH | noise-corr. | PSTH | noise-corr. |
| MLE + single-trial + NC | 0.91 ± 0.003 | 0.94 | 0.85 ± 0.004 | 0.96 |
| MLE | 0.90 ± 0.002 | 0.91 | 0.84 ± 0.003 | 0.96 |
| 2-step (CNN) | - | - | 0.87 ± 0.04 | 0.91 |
| 2-step | 0.72 ± 0.10 | 0.91 | 0.81 ± 0.05 | 0.95 |

Table S6: Performance comparison with the 2-step method [14] on the Retina Dataset. The performance of the 2-step dataset are taken from their paper. For this comparison we used their definition of $R^2$ which does not penalize a constant offset between prediction and target. For historical reasons, we used here a sample-and-measure NC loss with cross-entropy and not mean-square error. It works but considering later simulations on other datasets we expect higher $R^2$ with mean-square error.

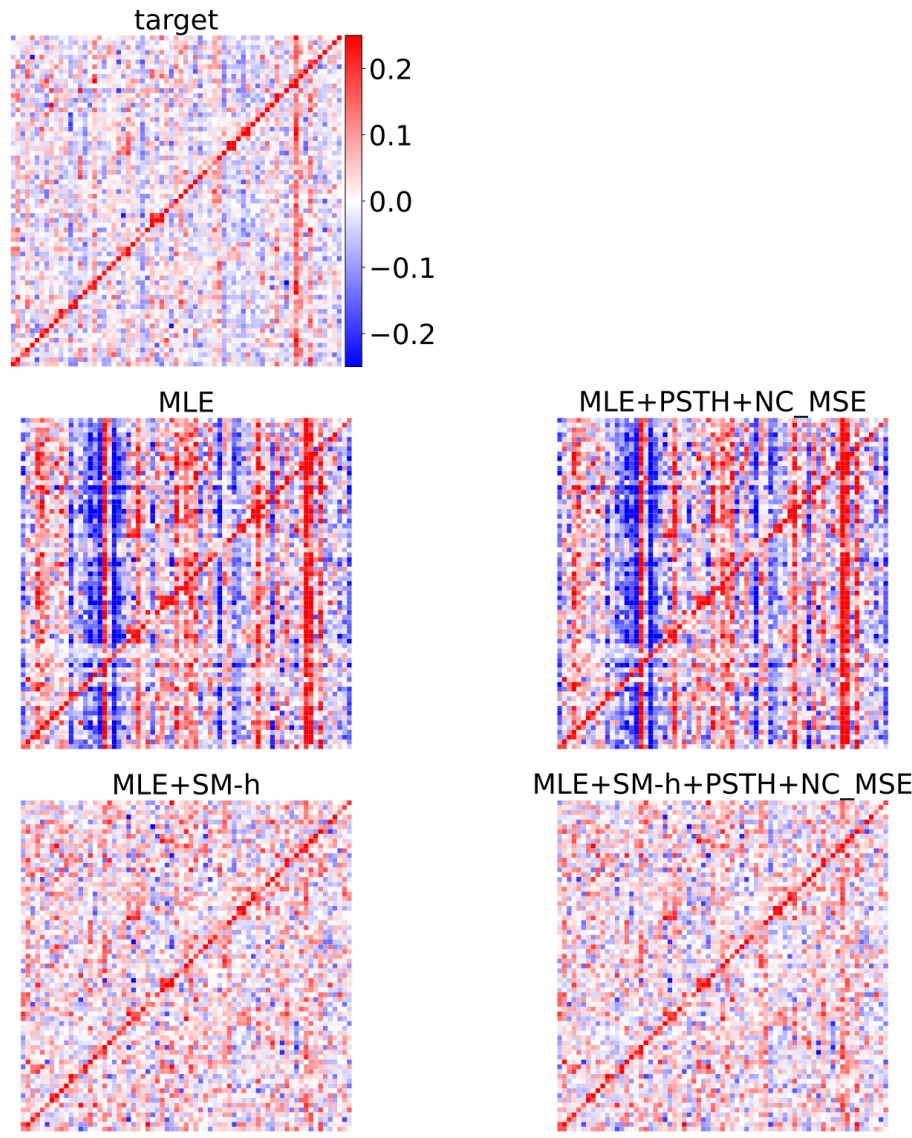

Figure S5: **High resolution plot of connectivity matrices in Figure 4 (large target network)**

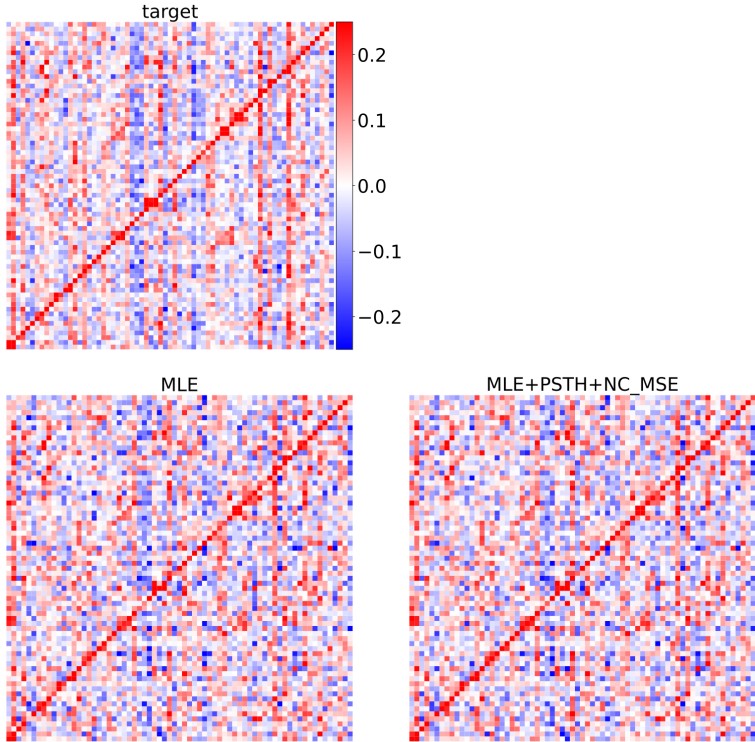

Figure S6: **High resolution plot of connectivity matrices in Figure 4 (small target network)**