# OpenReview forum: "Fitting summary statistics of neural data with a differentiable spiking network simulator"
_NeurIPS.cc/2021/Conference — NeurIPS 2021 Poster_

### Official Review · Reviewer_faLe · 2021-07-14

**Rating:** 6
**Confidence:** 3

**Summary:**

This study proposes an approach to fitting spiking networks to neural population data, by combining a typical maximum likelihood loss (in case of a fully observed network, but an ELBO loss for a network with unobserved neurons) with an additional loss term measuring the distance between summary statistics of recorded data and simulated data. This approach is shown to capture the summary statistics of interest better than a typical maximum likelihood approach (without this additional term) in a few scenarios.


**Limitations And Societal Impact:**

The authors do not sufficiently discuss the limitations of the study. In particular, the authors discuss the computational cost of the method as a function of the dimensionality of the fitted network, but do not include the performance of the method as a function of the dimensionality, which is arguably more important to report and discuss than the computational cost.



**Main Review:**

### Originality

The novelty of the work resides in the design and analysis of a new loss function for fitting spiking neural networks. Previous work is appropriately cited and it is clear how this work differs from previous contributions.


### Quality

The paper is technically sound, but some of the claims are not sufficiently backed by empirical evidence:

-the empirical results of Figures 3 and 4 seem to be for one run of the model. This is somewhat insufficient to assess the real difference in performance between losses, and increasing this to at least 10 repeats (e.g. different seeds) would be critical;

-the hidden neuron scenario of Figure 4C - where the neural activity is unobserved but the exact number of neurons is known - is quite unrealistic and I would recommend the authors to test a scenario where the exact number of hidden neurons is also unknown.


### Clarity

The manuscript is for the most part clearly written. However, it sometimes contains typos and imprecisions, such as:

-instead of "has been a fundamental tool to progress our understanding", "has been a fundamental tool to progress in our understanding";

-instead of "like the peri-stimulus histogram (PSTH)", "like the peri-stimulus time histogram (PSTH)";

-line 67, z is missing index k in the factorisation of the likelihood;

-line 83, \mu_{MLE} and \mu_{SM} are not defined (similarly, see line 152);

-"the visible spikes zV are clamped to the recorded data zD and the hidden spikes zH are sampled causally using our model". The term "causally" is slightly confusing. I would suggest the authors to either remove the term or rephrase the sentence;

-I would suggest the authors to improve the clarity of the sentence "Although minimizing LMLE recovers the true distribution PD under ideal conditions it does not seem to apply with neural data because the MLE solution often generates unrealistic activity";

-the sentence in lines 159-162 ("This interpretation...ideal for MLE") is not very rigorous and hard to parse, and would benefit from a rephrasing.



### Significance

The results suggest that the developed approach could be a promising step towards scalable methods for fitting spiking networks to neural population data. However, the study generally falls short of empirical evaluations of the method performance (note that the manuscript is one page less than the maximum, and would have the space for additional material).

### After rebuttal

Thank you for the answers to my review. Given the other reviews and the authors' rebuttal, and in particular the willingness of the authors to clarify the main contributions of the paper, I increased my rating to better reflect the quality and significance of the work.

**Time Spent Reviewing:**

3

---

> ### Author Response · Authors · 2021-08-10
> **Response to reviewer faLe**
>
> We would like to thank the reviewer. All comments about the clarity will be integrated in the final version of the manuscript. We will address the remaining comments point by point:
>
> * “the empirical results of Figures 3 and 4 seem to be for one run of the model. This is somewhat insufficient to assess the real difference in performance between losses, and increasing this to at least 10 repeats (e.g. different seeds) would be critical;”
>
> To address this comment within the short time available in the rebuttal period we have run 3 seeds for two simulations from Figure 3 and 5 seeds for 2 simulations from Figure 4C. In Figure we find that the negative log-likelihood of the MLE and MLE + PSTH + NC_MSE are respectively 0.37 +- 2e-5 and 0.37 +- 3e-4. For the noise-correlation matrix we find a R2 coefficient of -0.09 +- 0.07 and 0.77 +- 0.01. In Figure 4C, for the R2 of the connectivity matrix between visible neurons, we found the following mean and standard deviations: ELBO + SM-h 0.60+-0.007 and ELBO + SM-h + PSTH + NC 0.66+-0.008. Upon acceptance, we will provide standard deviations for all entries in these Figures.
>
> * “the hidden neuron scenario of Figure 4C - where the neural activity is unobserved but the exact number of neurons is known - is quite unrealistic and I would recommend the authors to test a scenario where the exact number of hidden neurons is also unknown.”
>
> We agree with the reviewer that this setting is not applicable as such with real data, and indeed, the scenario where the number of neurons is unknown is the most relevant. To show that our approach works in this scenario we ran additional experiments which extend Figure 4C and we suggest adding these results in the final manuscript. We have now fitted student networks with a varying number M of hidden neurons being 0, 10, 200 or 400. The validation loss happens to be minimized with M=200 but more interestingly, the R2 of the connectivity between visible neurons was respectively: -0.62, 0.51, 0.62, 0.66 (we ran a single seed for each configuration at the moment). When reporting these additional results we will state clearly that this enables the optimization of the number of hidden neurons but we do not see this as a method to guess the number of hidden neurons is the target network. Perhaps the most striking result here is the large difference of performance between M=0 and 10. In our eyes, it confirms the finding which was previously based on our results with 0 or 431 hidden neurons: if the hidden population is substantial in the target network it seems necessary to model this hidden activity to reconstruct the connectivity between visible units with a GLM. Selecting the right number of hidden neurons seems less critical. These new results will be added and discussed in the final manuscript.
>
> * In particular, the authors discuss the computational cost of the method as a function of the dimensionality of the fitted network, but do not include the performance of the method as a function of the dimensionality, which is arguably more important to report and discuss than the computational cost.”
>
> For the last experiment we clearly stated that this experiment is only “a prospective scaling experiment on a dataset from the Mouse Brain” to investigate how the method scales to very large networks. We agree that the fitted network as a function of the network dimensionality is very interesting but we decided to leave these analyses to future work because it requires to stitch data from multiple sessions and multiple brain areas which are hard problems going beyond the scope of this paper.

---

### Official Review · Reviewer_cqK7 · 2021-07-15

**Rating:** 5
**Confidence:** 5

**Summary:**

This work proposes to regularize the MLE with loss on statistics of model-simulated spike train to improve the estimation of recurrent spiking neuron network. It also introduces hidden neurons into the model in hope that it helps inference.

**Limitations And Societal Impact:**

The limitations are not adequately addressed. Comparisons with other work are needed.


**Main Review:**

Originality:  Previous work was discussed in the submission. However, it loses track of some recent work on the same topic.
- [Arribas et al. Neurips 2020] proposed similar idea to the sample-and-measure loss function. They optimize the maximum mean discrepancy (MMD) between data and simulated spike train or MMD-regularized MLE.
- Park et al. Neural computation 2012] proposed a general kernel-based framework that defines dissimilarity and divergence between spike trains.

Quality: Generally the submission shows theoretical and empirical analysis and evaluation to support the claim. However these properties lack rigorous proof and more evaluations are needed for comprehensive comparison.
- The model in this work looks more like a (autoregressive) GLM to me (It is called GLM in the citations e.g. [1]). Yet I will not dwell on arguing about the definition.
- MLE of such a model in the submission is a convex problem. However the sample-and-measure loss function is not necessarily convex w.r.t. the parameters in general. Then its existence and uniqueness of the global minimum is questionable. The authors should provide rigorous proof for these properties. Practically,  even if a global minimum exists, it is still unknown if it can be reached by the optimization algorithm used in this work.
- The V1 data used in the experiment is input-driven. It is also necessary to see the result of spontaneous activities, especially the long-term simulation.
- In real data, the recorded neurons always have input from other sources. The hidden neurons in the model could identify them, but it can hardly be enough for single neurons. It is more important to compare the model performance with/without hidden neurons for the same data.
- The empirical results only compared different losses on the proposed model. Comparisons with other work are needed.

Clarity: Overall the submission is clearly written and well organized. Please clarify the following
- How to determine the number of hidden neurons
- Choice of $\mu_{MLE}$ and $\mu_{SM}$
- Table 1: error of NC and NLL

Significance: One issue of MLE on such models with autoregressive components is that the model is actually different for training and simulation. During training the log-likelihood is conditional on spike history that is considered fixed, but the simulation is conditional on spike history that is generated randomly (random variables). This often results in unrealistic simulation like Fig.1E. The proposed method could help to deal with this issue.

**Time Spent Reviewing:**

10

---

> ### Author Response · Authors · 2021-08-10
> **Response to reviewer cqK7**
>
> We would like to thank the reviewer for pointing out the two references and they will be appropriately added in the related work section. We did not know the paper from Arribas and colleagues although it is interesting and relevant. This is similar to our work since it describes a method to fit GLMs and corrects for the instabilities of MLE, but there are fundamental differences in the way the gradients are computed through the free-running trajectories. Arribas and colleagues use score functions (referred to as the reinforce trick in our work) but they did not have backpropagation-like techniques to propagate gradient through the stochastically generated spikes. Multiple works [15,16,22] in machine learning have shown that training a network with score functions is less efficient than with backpropagation and we do not know other papers which present a way of propagate gradients through the stochastic spiking activity of GLMs.
>
> We will address the other comments from the reviewer about the paper quality point by point:
>
> * The model in this work looks more like a (autoregressive) GLM to me (It is called GLM in the citations e.g. [1]). Yet I will not dwell on arguing about the definition.
>
> We agree, this was specified in the introduction but we should repeat it in Section 2. We will clarify this in the final manuscript by replacing the sentence beginning at line 56 with the following:  “Similarly as in [1,2, 33, 12] we use a GLM where each unit is modeled as an SRM neuron where u_j^t …”. Note however, that our network model is different from the traditional GLM in two important ways: we use a CNN as a pre-processing step, and we consider hidden neurons. Figure S1 provides a comparison between our model and the official GLM matlab code when both models are trained with MLE: the PSTH correlation is improved with our model by a large margin.
>
> * MLE of such a model in the submission is a convex problem. However the sample-and-measure loss function is not necessarily convex w.r.t. the parameters in general. Then its existence and uniqueness of the global minimum is questionable. The authors should provide rigorous proof for these properties. Practically, even if a global minimum exists, it is still unknown if it can be reached by the optimization algorithm used in this work.
>
> We agree with the reviewer as long as there are no hidden neurons. In this simpler setting, the MLE is convex and our losses are not convex because our losses depend on spike trains generated by the model -- hence it's not convex like any loss applied to an RNN in deep-learning. So the solution is probably not unique but the existence is still guaranteed because the dissimilarity d is bounded from below. We suggest to acknowledge these in the manuscript. However, although our algorithm does not enjoy the theoretical guarantees of MLE, the solution we find is arguably better: it achieves a comparable likelihood and it generates a more realistic activity (see Table 1).
>
> Perhaps more importantly we would like to point that the presence of hidden neurons is central in our study, and in this setting the theoretical properties of MLE are compromised. With hidden neurons there are two problems: (1) the model is not identifiable and the MLE solution is not unique (one can for instance permute the identity of two hidden neurons) and (2) the likelihood is intractable and we have to replace it with a lower bound which is typically not convex. We will clarify in the final manuscript that one goal of Properties 1 and 2 was to provide properties of the sample-and-measure loss function which hold in the presence of hidden neurons where MLE is not applicable. They hold in the presence of hidden neurons because the identifiability of the statistical model is not required.
>
> Although we acknowledged above that our loss function is not convex when the data is finite. We would like to clarify that the uniqueness and the existence of the global minimum of $L_{SM+MLE}$ are guaranteed by Property 3 in the asymptotic limit. We therefore suggest to rephrase Property 3 as follows:
> >“If the RSNN is well specified and identifiable so that $P_D = P_{\phi *}$ and the data is infinite, then the global minimum of $L_{MLE+SM}$ exists, it is unique and equal to $P_{\phi *}$ .”
>
> We also suggest to clarify the proof accordingly:
> > “To prove this, we first note that all the conditions are met for the consistency of MLE so that $\phi *$ is the unique solution of $L_{MLE}$ . Also the assumption $P_D = P_{\phi *}$ is stronger than the assumption required in Proposition 1 and 2 so $P_{\phi *}$  is also the global minimum of $L_{SM}$ . As a consequence $\phi *$ is also the global minimum of the summed loss $L_{SM+MLE}$. This global minimum is also unique because it has to minimize $L_{MLE}$ which has a unique minimum .”
>
> Regarding the convergence of the training algorithm involving straight-through gradients to a local minimum, this is a hard problem which has not yet been proven (the closest result was the topic of the paper Yin 2019 https://arxiv.org/abs/1903.05662 but it does not apply in this stochastic setting). We have considered this to be slightly outside of the scope of this work. If the reviewer thinks that this is valuable we can still add a new intermediate result along with its proof:
> > "Each parameter update of stochastic gradient descent with our straight-through gradient estimate decreases strictly the sample-wise loss function if it is not already at a local minimum. If it is already at a local minimum the parameters remain unchanged."
>
> Although the loss function is decreasing sample after sample, it is harder to prove a form of convergence of the global loss function because of the stochasticity. In practice we see empirically that the training loss decreases steadily until a seemingly stationary value -- similar observations were reported in machine learning applications of straight through gradient estimates.
>
> * The V1 data used in the experiment is input-driven. It is also necessary to see the result of spontaneous activities, especially the long-term simulation.
>
> In the current manuscript we report results on three stimulus driven datasets from different labs and animals and one synthetic dataset, but we agree that it would be very interesting to study our method on a dataset of spontaneous activity. We did not have time to do that in the rebuttal period and we left this to future work.
>
> * In real data, the recorded neurons always have input from other sources. The hidden neurons in the model could identify them, but it can hardly be enough for single neurons. It is more important to compare the model performance with/without hidden neurons for the same data.
>
> We are not sure if we understood correctly what is meant here by “but it can hardly be enough for single neurons”. We believe that a relevant comparison with/without hidden neurons was performed on a synthetic dataset in the right columns of Figure 4B and 4C. The main difference there is the quality of the reconstruction of the connectivity between visible neurons: without hidden neurons it fails (Figure 4B bottom right) but it works with hidden neurons (Figure 4C bottom right). In these two panels, the data and the algorithms are otherwise the same.
>
> * The empirical results only compared different losses on the proposed model. Comparisons with other work are needed.
>
> We would appreciate having a more precise comment to know what additional comparisons are required. So far we had provided comparison with three other methods. Table S5 compares the sample-and-measure method with the 2-step method published in the same conference last year. Our method performs better on all the metrics except one where the performance is not really significant. Figure S2 provides a comparison with the SpikeGan approach as discussed in the main text, Figure S1 compares with the classical GLM-MLE approach and the official implementation.
>
> * How to determine the number of hidden neurons
>
> We agree with the reviewer that this is an interesting question, in particular because we only tried to fit a student network in Figure 4C which has the same number of neuron as the target network. This may appear to be too easy. Hence we would like to address this question by adding new results in the final manuscript. We have led additional experiments which extend the right columns of Figures 4B and 4C. We ran simulations with a varying number of hidden neurons M being 0, 10, 200, or 400 (the target network had 431 hidden neurons). We find in our case that the validation loss is minimal with 400 hidden neurons but we cannot exclude that a larger hidden population size would be better. We also found coefficients R2 for the reconstructed connectivity between visible neurons to be:  -0.62, 0.51, 0.62, 0.66 respectively. In our eyes, the most striking result here is that the largest difference of R2 happens between M=0 and 10. It suggests that modeling the hidden activity is very much necessary but choosing the right number of hidden neurons is less critical. These new experiments will be discussed accordingly in the final version of the manuscript to support this point more strongly.

---

### Official Review · Reviewer_nRQi · 2021-07-16

**Rating:** 7
**Confidence:** 4

**Summary:**

This paper introduces a novel approach to fitting spiking network
models to neural data. The proposed method uses differentiable
recurrent spiking networks, which are fit to the data by minimising a
loss function with the help of recent techniques such as the
straight-through gradient estimate. The key novelty is to generalise
the loss function to be minimized, to include not only the likelihood
of the data, but also a number of regularizing terms that depend on
desired spike train statistics, such as the PSTH or the noise
correlations. These domain-specific terms in the loss function help
with breaking degeneracies in the model, and with improving fit
quality in the face of model misspecification. After a theoretical
analysis of the properties of the proposed loss function(s), the paper
shows examples of applications of the technique to some simulated
data. A comparison to a state-of-the-art generative model for
population spike trains, and an example of usage of the proposed
method on a large database of experimental recordings, are presented
in the supplementary material.

**Limitations And Societal Impact:**

The authors have adequately addressed the limitations and potential negative societal impact of their work.

**Main Review:**

In my opinion, the paper brings an interesting perspective to a topic
that is of sufficiently broad interest within the NeurIPS community, and
the idea behind the proposed technique is, as far as I am aware, a
novel combination of known methods. The effectiveness of the
sample-and-measure method in improving fit quality, especially in the
face of model misspecification, is convincingly presented, and I
enjoyed the geometrical interpretation of the technique. the authors
do a good job of comparing with another relevant method from the
recent literature, and of giving the reader an idea of how the
technique could scale up to large datasets, which are becoming more
and more common. The paper is clearly written, and I was not able to
spot any technical issue or shortcoming. Overall, I think this is a
solid contribution.

My only critique to the work is that the claims on the usefulness of
this method for connectivity reconstruction seem not to be very
strongly supported by the evidence presented. More specifically, the
claim that "the sample-and-measure method [...] is important to
reconstruct trustworthy connectivity matrices in cortical areas"
(lines 284--288) rests upon the fact that a correctly-specified model
with hidden neurons can recover the connectivity matrix of the visible
population with an R² "almost as high" (line 281) as in the case where
the full population is visible. I have two objections to this:
1. the R² for the proposed method with hidden neurons is 0.64, while
   in the fully-visible case it is 0.66. For the baseline MLE method,
   these numbers are 0.59 vs 0.65. In an absolute sense the numbers
   don't look that far off, and in absence of systematic statistical
   assesments it is hard to tell if the difference between the two
   methods is really up to the difference between them, or it depends
   on the specific dataset/network being simulated.
2. even taking the claim at face value, what the data in figure 4c
   show is that connectivity can be recovered in presence of hidden
   neurons (whether better or not than by simple MLE is up for debate,
   see previous point), *if we work with a perfectly well-specified
   model*. Indeed, and unsurprisingly, the only results we have for a
   misspecified model (figure 4b) show that, even with the
   sample-and-measure method, connectivity estimates should not be
   trusted. But in practice, with real data, our models will be always
   misspecified, so it is not clear how useful it is to show that
   connectivity can be recovered in the well-specified case.
To summarise my two points above, I think that the good performance of
the model at recovering the connectivity can be explained mostly by
the fact that the model is well specified (and indeed, MLE does pretty
well too), rather than as a specific property of the proposed method.


Minor points and typos
- Line 220: "instead of a CNN" (remove "the").



**Time Spent Reviewing:**

3

---

> ### Author Response · Authors · 2021-08-09
> **Response to reviewer nRQi**
>
> Thank you for the thorough review of our paper and the positive feedback. The reviewer raises two objections that we address below:
>
> 1. the R² for the proposed method with hidden neurons is 0.64, while in the fully-visible case it is 0.66. For the baseline MLE method, these numbers are 0.59 vs 0.65. In an absolute sense the numbers don't look that far off, and in absence of systematic statistical assesments it is hard to tell if the difference between the two methods is really up to the difference between them, or it depends on the specific dataset/network being simulated.”
>
> We would like to point out that the competing method ELBO + SM-h which “does look that far off” was described in this paper for the first time and benefited from multiple innovations of the present paper: (1) the ELBO loss is not canonical and it optimized with straight-through gradient for stochastic RSNNs; (2) the SM-h components refers to a regularization of the average firing rate of the hidden units. We found in fact that this regularization is crucial for the stability of this ELBO loss (we will add a supplementary Figure to support this). In this dataset the only algorithm which was published elsewhere is MLE and it fails completely in this setting (see Figure 4B and 4C), for a comparison with a more contemporary algorithm we refer for instance to the lines 208 - 223 or Table S5. Note also that we re-ran the fitting algorithm with 5 random seeds and found that the ablated algorithm ELBO + SM-h reliably reaches a significantly lower R2 (0.60 +- 0.007) than the full algorithm ELBO + SM-h + PSTH + NC (0.66 +- 0.008).
>
> 2. even taking the claim at face value, what the data in figure 4c show is that connectivity can be recovered in presence of hidden neurons (whether better or not than by simple MLE is up for debate, see previous point), if we work with a perfectly well-specified model. Indeed, and unsurprisingly, the only results we have for a misspecified model (figure 4b) show that, even with the sample-and-measure method, connectivity estimates should not be trusted. But in practice, with real data, our models will always be misspecified, so it is not clear how useful it is to show that connectivity can be recovered in the well-specified case. To summarise my two points above, I think that the good performance of the model at recovering the connectivity can be explained mostly by the fact that the model is well specified (and indeed, MLE does pretty well too), rather than as a specific property of the proposed method.
>
> The reviewer worries here that the successful results of Figure 4C could emerge because the model is well specified. It is true indeed that the student network in Figure 4C has exactly 431 neurons which happens to be the number of neurons in the target network. We would argue however that our method does not need the number of hidden neurons to be correctly specified as long as a substantial hidden population is modeled.
>
> To demonstrate this we have run additional experiments and we will add a new Figure in the final version of the manuscript. We fitted a student network using the full sample-and-measure method ELBO + SM-h + PSTH + NC_MSE as in the right column of Figure 4C but with M hidden units where M is either 0, 10, 200, or 400. For these four different numbers of hidden units we recover the connectivity between visible units with coefficients of determination R2: -0.62, 0.51, 0.62, 0.66. Note that when M equals 0 or 431 it comes back to the case in the right columns of Figure 4B/ or 4C. Although adding hidden neurons in the student network tends to improve the performance, all solution with M > 0 already work much better than the MLE solution from Figure 4B. In fact the largest gap is between M=0 and 10, suggesting that the most important ingredient is to model some hidden neurons, but the exact number matters less.  These new results will be added to the manuscript and discussed accordingly.

---

> > ### Comment · Reviewer_nRQi · 2021-08-31
> > **response to rebuttal**
> >
> > I wish to thank the authors for their response - I am satisfied by how my concerns were addressed. I had missed a point that can be summarized by the following lines in the authors' response:
> >
> > > the competing method ELBO + SM-h which “does look that far off” was described in this paper for the first time and benefited from multiple innovations of the present paper
> >
> > and
> >
> > > In this dataset the only algorithm which was published elsewhere is MLE and it fails completely in this setting (see Figure 4B and 4C)
> >
> > Given the above, my suggestion to the authors is to shift some emphasis away from the PSTH+NC results in the text, towards better underscoring the merits that the ELBO+SM-h method (with the innovations introduced in this paper) already possesses by itself.
> >
> > I confirm my initial score.

---

### Official Review · Reviewer_k9D3 · 2021-07-18

**Rating:** 5
**Confidence:** 3

**Summary:**

This paper proposes a new method to fit a recurrent spiking neural network (RNN) to neural data, which consists in simultaneously maximizing the likelihood of the spike trains observed (MLE), while fitting other observables such as averaged activity over trials (PSTH) and noise correlations (NC). On real neural data (monkey V1, rat retina), the method is shown to better capture these other observables compared to models only trained on MLE. On semi-synthetic data (artificial network fit to match the V1 dataset with additional hidden neurons), the method is shown to slightly better reflect the underlying connectivity profile compared to a method trained only on MLE.

**Ethical Concerns:**

No.

**Limitations And Societal Impact:**

Yes.

**Main Review:**

__Summary of the review:__ The proposed optimization procedure to fit RNNs to neural data is sophisticated and theoretically well-justified. However the quantitative results of the method are not entirely convincing, the potential of the method to bring new insights to neuroscience is not clearly demonstrated, and the work would greatly benefit from being better tied to prior work. For these reasons, I am not fully convinced that this paper is ready for acceptance at NeurIPS in its current state.

__Strengths__
- The optimization procedure proposed is sophisticated, thoroughly described and well-justified.
- The theoretical insight that fitting these additional observables can be seen as a Bayesian prior on the space of solutions is clear and interesting.

__Limitations__
- The quantitative benefits of the method are not entirely convincing. On real data, the method is shown to indeed improve the fit of the observables compared to a pure MLE model, which is expected, but perhaps disappointingly the MLE fit itself in not improved by the method. On synthetic data, the network connectivity (R2 = 0.64) recovered by fitting PSTH and NC is barely improved compared to a model which does not fit these additional observables (R2 = 0.59).
- The potential of this method to bring new insights to neuroscience is not clearly demonstrated.
- The paper would benefit greatly from being better connected to prior art:

LFADS is a prior effort of fitting RNN to neural data (it would be nice to understand the difference with this work):

*LFADS - Latent Factor Analysis via Dynamical Systems
David Sussillo, Rafal Jozefowicz, L. F. Abbott, Chethan Pandarinath
https://arxiv.org/abs/1608.06315*

Previous work has shown that the underlying connectivity of a network is very difficult to estimate from the neural activity:

*Systematic errors in connectivity inferred from activity in strongly coupled recurrent circuits
Abhranil Das,  Ila R. Fiete
https://www.biorxiv.org/content/10.1101/512053v1*

There has been a lot of efforts in fitting various observables of population statistics, which are not discussed in the paper. For a review:

*Modeling the correlated activity of neural populations: A review
https://arxiv.org/abs/1806.08167*

For a specific example: the K-pairwise model is the maximum entropy model reproducing the mean spiking activities,
pairwise correlations, population count histogram P(K):

*Tkacik, G., Marre, O., Amodei, D., Schneidman, E., Bialek,
W., and Berry II, M. J. (2014). Searching for Collective
Behavior in a Large Network of Sensory Neurons. PLoS
Comput Biol, 10(1), e1003408.*

*Tkacik, G., Marre, O., Mora, T., Amodei, D., Berry II,
M. J., and Bialek, W. (2013). The simplest maximum entropy model for collective behavior in a neural network.
Journal of Statistical Mechanics: Theory and Experiment,
2013(03), P03011.*

On fitting sequential recordings of overlapping populations (a goal mentioned in the Perspective section):

*Oleksandr Sorochynskyi, Stéphane Deny, Olivier Marre, and Ulisse Ferrari. Predicting synchronous firing of large neural populations from sequential recordings. arXiv preprint arXiv:1904.04544, 2019.*

__Minor__
- It would be interesting to see a fit of the PSTHs in the plots.

**Time Spent Reviewing:**

3

---

> ### Author Response · Authors · 2021-08-09
> **Response to reviewer k9D3**
>
> We are grateful for the careful reading of our paper by the reviewer. We address the limitations raised by the reviewer point by point:
>
> * “The quantitative benefits of the method are not entirely convincing. On real data, the method is shown to indeed improve the fit of the observables compared to a pure MLE model, which is expected, but perhaps disappointingly the MLE fit itself in not improved by the method. On synthetic data, the network connectivity (R2 = 0.64) recovered by fitting PSTH and NC is barely improved compared to a model which does not fit these additional observables (R2 = 0.59).”
>
> On real-data, it is true that our network solution does not provide a higher likelihood than the MLE solution. Figure 1C explains why this is the case: the likelihood landscape is flat around the MLE so that many network solutions are equally good in terms of likelihood. For instance the MLE and MLE + PSTH solution from Figure 3 B and C reach similar log-likelihoods (as indicated by the reviewer) even if the MLE solution produces very unrealistic activity. Hence the likelihood should not be the decisive metric to judge the quality of the fitted model. On other metrics we are better that the MLE solution (see Table 1). We also improve upon other methods like the 2-step method published in the same conference last year (see Table S5).
>
> On synthetic data, the reviewer points out that differences between the model ELBO + SM-h and the full model ELBO + SM-h + PSTH + NC are small. We would like to point out that the ELBO + SM-h baseline uses a number of innovations of the present paper, namely: (1) we optimize a custom likelihood lower-bound which can be optimized with straight-through gradients (2) without regularization of the hidden firing rates (SM-h) the ELBO solution alone is not stable (we suggest adding a supplementary figure to support this point in the appendix). In fact we do not know any published methods which could optimize GLM networks of this size with hidden units. This algorithm in Figure 4C was intended as an intermediate result to understand which component of the algorithm contributed the most. We will make this clear in the final version of the manuscript. Moreover, after running the fitting procedure with five random seeds we find that the R2 of ELBO + SM-h (0.60 +- 0.007) is significantly inferior to the R2 obtained with ELBO + SM-h + PSTH + NC (0.66 +- 0.008). We will add the standard deviations in the final manuscript.
>
> Overall in Figure 4, the only baseline algorithm which was previously described is the MLE solution which ignores the hidden neurons from the left column of Figure 4B, and it completely fails on this dataset: it achieves a negative R2 when the target network has hidden neurons but the student network does not. For a quantitative comparison with a more recent algorithm, we have reported in Table S5 that our method improves upon the 2-step method from NeurIPS 2020 on almost all metrics (only the difference in PSTH correlation obtained on the checkerboard stimulus is not clearly significant).
>
> * “The potential of this method to bring new insights to neuroscience is not clearly demonstrated.”
>
> We would like to argue against this point of view. Fitting MLE to neural data has been used in multiple high-impact neuroscience papers in the last 15 years (references [1,2,3,4,5,6,7,8]). We have shown here that this traditional method, MLE, often generates unrealistic activity statistics which is fully corrected with our sample-and-measure algorithm. We also identify that the GLM-MLE approach fails to reconstruct the network connectivity when some neurons are not observed (negative R2 in Figure 4B, left). This will always be the case for cortical recording which are likely to get many external inputs, and we show that our method can mitigate this issue (Figure 4C, we do not know any other method capable of doing this). We believe that our method makes a significant progress towards reliable methods for neural circuit reconstruction.
>
> * The paper would benefit greatly from being better connected to prior art:
>
> Thank you for the relevant references, note however that other reviewers have written positive feedbacks about the way we reported prior works:
>
> > Reviewer nRQi: “the authors do a good job of comparing with another relevant method from the recent literature”
> > Reviewer cqK7 “Previous work was discussed in the submission.” (this reviewer also added  two references which are different from those discussed below) .
> > Reviewer faLe: “Previous work is appropriately cited and it is clear how this work differs from previous contributions.”
>
> But still, the pointers added by the reviewer are relevant, we will add them appropriately in our paper and we discuss the relationship with our work below:
>
> > LFADS by Sussillo and colleagues:
>
> In this work a bi-directional RNN is used to express an underlying latent variable from which the spiking activity can be regenerated. It models more abstractly the network computation than our approach because (i) information can flow in the network model both forward and backward in time (ii) the recurrent dynamics are not spiking (iii) a unit of the RNN in this work is not modeling a single recorded neuron but a latent variable. Although these approaches can be powerful generative models of neural activity (which deserves a citation in our manuscript) they cannot model the underlying network connectivities or the intrinsic neural dynamics as done with GLMs [4,5,6,7,8].
>
> > “Previous work has shown that the underlying connectivity of a network is very difficult to estimate from the neural activity:
> > Systematic errors in connectivity inferred from activity in strongly coupled recurrent circuits Abhranil Das, Ila R. Fiete.”
>
> Thank you very much for this pointer. Indeed, this paper warns the community about the difficulty of recovering neural connectivity. We will add this reference to stress that recovering the network connectivity is very hard and the link between the recovered connectivity and the reconstructed connectivity should always be taken with a grain of salt. However this paper also attempts to retrieve the connectivity of a synthetic network like we do in Figure 4. Das and Fiete much like reference [5] have shown that GLMs are quite successful on such artificial data. Our paper going one step further as we demonstrate that the classical GLM-MLE approach fails drastically if there are hidden neurons which are ignored in the student network. We also provide algorithm which can model the hidden activity and solve this problem.
>
> > “There has been a lot of efforts in fitting various observables of population statistics, which are not discussed in the paper. For a review:
> > Modeling the correlated activity of neural populations: A review https://arxiv.org/abs/1806.08167
>
> > For a specific example: the K-pairwise model is the maximum entropy model reproducing the mean spiking activities, pairwise correlations, population count histogram P(K):
>
> > Tkacik, G., Marre, O., Amodei, D., Schneidman, E., Bialek, W., and Berry II, M. J. (2014). Searching for Collective Behavior in a Large Network of Sensory Neurons. PLoS Comput Biol, 10(1), e1003408.
>
> > Tkacik, G., Marre, O., Mora, T., Amodei, D., Berry II, M. J., and Bialek, W. (2013). The simplest maximum entropy model for collective behavior in a neural network. Journal of Statistical Mechanics: Theory and Experiment, 2013(03), P03011.”
>
> We will add all three references. We agree that this literature is interesting and provide other solutions to fit activity statistics. It is interesting to mention methods relying on Ising models rather than GLMs. Note however that reference [12] is a more recent paper from some of these authors, this paper is central in our work and we compared our results in detail on a dataset from one of their labs in Table S5.
>
> > Oleksandr Sorochynskyi, Stéphane Deny, Olivier Marre, and Ulisse Ferrari. Predicting synchronous firing of large neural populations from sequential recordings. arXiv preprint arXiv:1904.04544, 2019.
>
> Thank you, we will add this reference in the perspective section.
>
> > It would be interesting to see a fit of the PSTHs in the plots.
>
> We will add one such plot in the appendix.

---

> > ### Comment · Reviewer_k9D3 · 2021-08-30
> > **Thank you for rebuttal - Question about the main contribution to neuroscience and framing of the paper**
> >
> > Thank you for your rebuttal which clarifies some of my concerns.
> >
> > If there is still time (my understanding is that the deadline for discussions is September 2), I would like to ask what the authors think the main contribution of the paper to neuroscience is, and whether the paper could be re-framed to make this clearer.
> >
> > Is the main contribution of the paper the ability of the model proposed to infer the connectivity of an artificial network with unobserved units? If yes, it seems only weakly related to the main method described. I agree that ELBO + SM-h baseline uses a number of innovations of the paper, but I would argue that the main method described is mostly about fitting the additional variables PSTH + NC, which does not prove very useful even for this synthetic dataset.
> >
> > If we let aside the connectivity reconstruction contribution, and consider the GLM-fit contribution, I would like to understand better the importance of this contribution. Especially:
> > - What are the technical difficulties of fitting PSTH and NC in addition to MLE that this paper overcomes? The paper uses a straight-through gradients operators, which is known from prior work [15,16, 17], and a sample-and-measure strategy to sample spike trains and measure the dissimilarity between the recorded and simulated data, which seems like the straight-forward and logical thing to do.
> > - what new insights could this new model bring to neuroscience? It is expected that a model that fits PSTH and NC in addition to MLE would reflect PSTH and NC better, but the question that I think is important to answer is: what can a neuroscientist use such a model for?
> >
> > Thank you for your help clarifying the contributions of the paper.

---

> > > ### Author Response · Authors · 2021-08-31
> > > **Response from the authors**
> > >
> > > Thank you very much for acknowledging our reply, and the clarifications that we had written in there. Regarding the remaining questions:
> > >
> > > > What are the technical difficulties of fitting PSTH and NC in addition to MLE that this paper overcomes? The paper uses a straight-through gradients operators, which is known from prior work [15,16, 17] and a sample-and-measure strategy to sample spike trains and measure the dissimilarity between the recorded and simulated data, which seems like the straight-forward and logical thing to do.
> > >
> > > We agree that this is the straight-forward and logical thing to do, nonetheless we do not know any paper which has described a way to back-propagate through the simulated dynamics of GLM networks or any other type of stochastic recurrent spiking neural network. Technically, although the straight-through approach is well studied for feedforward networks of Bernoulli neurons or networks with deterministic discrete units, it was unclear if it also works for recurrent networks of stochastic spiking neurons without producing exploding or vanishing approximate gradients. With our choices of the parametrization and the pseudo-derivative we show that it works in this setting and even in the more difficult one which includes hidden neurons.
> > >
> > > Perhaps more importantly, the formalization of the loss function is also a relevant technical contribution for neuroscience because we show that it stabilizes the MLE on datasets of limited size acquired in neuroscience, with the additional theoretical guarantee that it would not bias the solution away from the MLE when the recorded data becomes sufficient. In this sense we provide a natural and provably correct way of improving functional network reconstruction by fitting summary statistics, which a logical thing to do as indicated by the reviewer.
> > >
> > > > what new insights could this new model bring to neuroscience? It is expected that a model that fits PSTH and NC in addition to MLE would reflect PSTH and NC better, but the question that I think is important to answer is: what can a neuroscientist use such a model for?
> > >
> > > The PSTH and NC are statistics which are thought to capture some of the "function" of the neurons. For instance stimulus selectivity, tuning curves, and more generally spatio-temporal receptive fields are part of our concept of PSTH since our approach can deal with PSTH responses given the stimulus, i.e., tuning as a function of the stimulus is a special case. Our work shows that standard MLE alone provides worse fitting accuracy for these statistics suggesting that a lot of the function of the reconstructed network is potentially compromised with MLE but recovered with our approach.
> > >
> > > To name a few concrete scientific studies which could benefit from our work. (i) Neural responses to visual stimuli by fitting spiking neurons:  we expect that improving PSTH fitting accuracy could improve the analysis in reference [a] and related papers. (ii) Coupling between areas:  Improving the accuracy of noise correlations using networks reconstructed with recurrent GLMs is expected to provide an advance against standard GLM in combination with MLE as used to study the interactions between prefrontal cortex and mediodorsal thalamus [3] or between auditory cortex and posterior parietal cortex [2]. There are also other simple statistics characterizing the distribution of inter-spike intervals of individual neurons which are likely to complement MLE efficiently when one is interested in the intrinsic dynamics and time constants of a individual neurons like in papers [6,7,8].
> > >
> > > We agree that these type of application of our method have not been described in our current manuscript and we suggest to add in the final version of the manuscript a paragraph (along the lines sketched here) in the discussion to address this.
> > >
> > > > Is the main contribution of the paper the ability of the model proposed to infer the connectivity of an artificial network with unobserved units?
> > >
> > > No, inference of connectivity is only one of many potential applications. We agree that our manuscript put too much weight on connectivity reconstruction than it should have done. Connectivity reconstruction is indeed a very difficult problem. Although we show that we improve upon existing techniques, the problem is far from solved. In the final version of the manuscript we will put less emphasis on connectivity reconstruction in the abstract, introduction and conclusion and discuss more the other applications of GLM networks as discussed above.
> > >
> > > [a] Stéphane Deny, Ulisse Ferrari, Emilie Macé, Pierre Yger, Romain Caplette, Serge Picaud, Gašper Tkačik, and Olivier Marre
> > > Multiplexed computations in retinal ganglion cells of a single type
> > > Nature Communications 2017

---

### Decision · Program_Chairs · 2021-09-27

**Decision:**

Accept (Poster)

**Comment:**

Dear authors,

Congratulations on your paper being accepted at Neurips. Your submission was discussed extensively, (and, at times, controversially) amongst the reviewers. We appreciated that your approach has the potentially to substantially increase the usability of a common analysis tool in neuroscience, by ensuring that GLMs fitted to multi-neuron spike recordings lead more faithful models of neural firing. The current ’state of the art’ is still to use MLE, and there is mounting empirical evidence that this often yields models that do not provide accurate generative samples (e.g. exploding firing rates, or inaccurate representations of PSTHs and correlations). The approach to extend the loss function and to use differential simulators for optimisation seems to provide a clear path for overcoming this general  problem — therefore, the final decision was to accept this paper at Neurips.

At the same time, the discussion and review process revealed multiple points in which the paper could be strengthed— please see the reviews and comments by the reviewers for specific points. However, a general theme relates to the framing of the paper— it is clear that approach leads to better generative samples, and this is demonstrated empirically. It is not immediately clear why this approach would also lead to better reconstructions of 'functional connectivity’ (aside from the many issues plaguing this concept), and the empirical evidence is similarly not completely convincing.

We would ask the authors to take the feedback by the reviewers seriously, and adjust the paper accordingly.

Best,

Your AC